# PDL2$^+$ CD11b$^+$ dermal dendritic cells capture topical antigen through hair follicles to prime LAP$^+$ Tregs

Leticia Tordesillas[1,2], Daniel Lozano-Ojalvo[1,2], David Dunkin[3], Lucie Mondoulet[4], Judith Agudo[2], Miriam Merad[2], Hugh A. Sampson[1,2,4] & M. Cecilia Berin[1,2]

The skin immune system must discriminate between innocuous antigens and pathogens. Antigen applied topically using a Viaskin® patch elicits immune tolerance that can suppress colitis and food allergy. Here we show how topical antigen is acquired and presented by dendritic cells in the skin. Topical antigen is acquired by Langerhans cells (LC) and CD11b$^+$ cDC2s but not cDC1s, and both LCs and CD11b$^+$ cDC2s reaching the lymph node can prime T cells and expand LAP$^+$ Tregs. However, LCs are neither required nor sufficient for T cell priming, and have no role in tolerance induction. Conversely, IRF-4-dependent cDC2s are required for T cell priming. Acquisition of antigen in the dermis, delivery to the draining lymph node, and generation of tolerance are all absent in hairless mice. These results indicate an important function for hair follicle niche and CD11b$^+$ cDC2s in antigen acquisition, and in generation of primary immune tolerance to topical antigens.

---

[1] Pediatric Allergy & Immunology, Icahn School of Medicine at Mount Sinai, New York 10029 NY, USA. [2] Immunology Institute. Icahn School of Medicine at Mount Sinai, New York 10029 NY, USA. [3] Pediatric Gastroenterology, Icahn School of Medicine at Mount Sinai, New York, NY 10029, USA. [4] DBV Technologies, Montrouge 90120, France. Correspondence and requests for materials should be addressed to M.C.B. (email: cecilia.berin@mssm.edu)

The skin, like other barrier sites, is an immunologically active organ that must discriminate between potentially harmful pathogens and innocuous antigens. Antigen is acquired and presented by dendritic cells, which include Langerhans cells (LCs) in the superficial epidermal layer and several dendritic cell subsets (DCs) in the dermis. Antigen applied topically can elicit host protective immunity, allergy, or immune tolerance depending on the context of antigen exposure[1–6]. DCs carry antigen acquired in peripheral tissues to draining lymph nodes, where they are essential for the priming of naïve T cells. The nature of the T cell response is determined by the context of antigen presentation, and one hypothesis to explain the heterogeneity of the immune response to topical antigen is that subsets of DCs are specialized for the induction of immunity, allergy or tolerance[7].

DCs can be divided into subsets based on ontogeny and/or expression of surface markers. Unlike DCs, LCs are independent of the growth factor Flt3L and share differentiation pathways with macrophages[8]. Classical DCs (cDCs) in the dermis can be divided into cDC1 and cDC2 subsets based on their dependence on IRF8/Batf3 and IRF4, respectively[9]. cDC1 and cDC2 subsets in the skin can be loosely divided based on expression of CD103 and CD11b, respectively, although there is also a population of CD103−CD11b− DCs that are IRF4 dependent. Functional specialization of these two subsets has been proposed, with cDC1 better able to induce CD8 T cell and Th1 responses for optimal response to intracellular pathogens[10,11], and cDC2 better able to induce Th2 and Th17 responses to respond to extracellular pathogens[12,13]. Surface expression of PDL2 or CD301b on CD11b+ cDC2 has been associated with Th2-priming capacity[12,14]. Regulatory responses have also been ascribed to different subsets of DCs, including CD11b+ cDC2s that express high levels of RALDH[15], and langerin+ dermal DCs and LCs[16–18]. However it is possible that presentation by any DC subset in the absence of adjuvant can lead to regulatory T cells (Tregs) and immune tolerance.

We have previously shown that topical application of antigen to intact skin with a Viaskin patch can generate immune tolerance that can suppress delayed-type hypersensitivity (DTH) responses, food allergy and inflammatory bowel disease[4,5]. Topical application of antigen generated antigen-specific LAP+ Foxp3− Tregs that expressed CCR9 and CCR6 to support intestinal homing, and suppressed T cell and mast cell activation through TGFβ dependent mechanisms[4,5]. These cells are similar in phenotype to Th3 cells identified as playing a critical role in the development of oral tolerance[19–21]. LAP+Foxp3− Tregs have also been shown to play a role in suppression of allergic inflammation of the lungs[22].

To determine how antigen applied topically to healthy skin is acquired and presented by skin DC subsets to generate LAP+ Tregs, here we show that LCs and CD11b+ cDC2s acquire and present topical antigen to T cells, but only cDC2s are required for T cell priming. Antigen acquisition and generation of tolerance are absent in hairless mice, suggesting a key role of hair follicle niche in delivery of topical antigen to skin DCs.

## Results

**Topical antigen is transported by CD11b+ cDC2s and LCs.** We examined the acquisition of ovalbumin (OVA) by DCs of the epidermis and dermis using Viaskin® patches loaded with OVA-AlexaFluor 647 ($OVA^{AF647}$). The gating strategy is shown in Supplementary Figure 1. The skin of Balb/c mice was prepared by removing the hair using clippers and depilatory cream one day prior, as previously described[4,5]. OVA was readily detectable in CD11c+ MHCII+ cells in the epidermis and dermis (Fig. 1a),

and kinetic analysis between 12 and 72 h after patch application showed a peak at 12 h that declined thereafter. Analysis of other cell types acquiring OVA in the dermis showed that macrophages (CD11b+ CD64+) acquired most of the antigen reaching the dermis (Fig. 1b). In the dermis, several DC subsets could be identified based on CD103, EpCAM, and CD11b. The $OVA^{AF647+}$ CD11c+ MHCII+ population in the dermis was predominantly but not exclusively of the cDC2 phenotype, expressing CD11b but not EpCAM (Fig. 1c).

We next examined transport of $OVA^{AF647}$ to the draining lymph node (LN). We detected $OVA^{AF647+}$ DCs in the skin-draining LN as soon as 12 h after applying the patch (Fig. 2a, b) with maximum antigen detected at 24 h. $OVA^{AF647+}$ DCs were detected at early time points (12−48 h) only in the skin-draining brachial LN but not in distal inguinal or mesenteric LN (Fig. 2c), although a small signal in distal LNs was detected after 72 h. $OVA^{AF647+}$ cells were predominantly CD11c+ MHCII$^{high}$ cells, corresponding to migratory DCs, although some antigen uptake was also observed in CD11b+ CD64+ macrophages (Fig. 2d). Antigen delivery to the lymph nodes, but not uptake by DCs or macrophages in the dermis, was dependent on CCR7 (Fig. 2e, f).

Using the surface markers CD103, EpCAM, CD11b and CD8a in addition to CD11c and MHCII, we could distinguish 5 DC subsets in the draining LN (Fig. 2g). Cells that were CD11c+ MHCII+, and $OVA^{AF647+}$ were present in two DC subsets: CD103−EpCAM+ CD11b$^{int}$ (that were named EpCAM+) DCs and CD103-EpCAM−CD11b+ (named CD11b+) DCs (Fig. 2g). These markers are consistent with LCs and cDC2s. CD103+, CD8+, and double-negative (Neg) DCs did not acquire topical antigen. $OVA^{AF647+}$ CD11b+ DCs migrated faster than $OVA^{AF647+}$ EpCAM+ DCs, and were first detectable after 12 h while EpCAM+ DCs were only detected 48 h after antigen application (Fig. 2g).

**EpCAM+ and CD11b+ DCs present topical antigen to T cells.** To study the capacity of DC subsets in the LN to present captured antigen, we purified DCs from the LN 48 h after topical application and co-cultured cells with CD4+ T cells purified from DO11.10 mice. DCs purified from the LN of OVA exposed mice induced specific proliferation of DO11.10 T cells that was abolished in the presence of anti-MHCII antibodies (Fig. 3a). As we previously demonstrated in vivo after topical application of OVA[4], OVA-loaded skin-draining DCs induced an expansion of LAP+ Tregs in vitro (Fig. 3a). DC subsets were sorted 48 h after patch placement, following the same gating strategy as described in Fig. 2g, and co-cultured with DO11.10 CD4+ T cells. EpCAM+ DCs and CD11b+ DCs, but not CD103+, CD8+ or Neg DCs were able to induce proliferation in responder T cells (Fig. 3b). In vitro loading of all DC subsets with OVA peptide as positive control led to proliferation, with the exception of CD8+ DCs (Supplementary Table 1). A similar proliferative response and induction of LAP+ T cells was noted in response to CD11b+ and EpCAM+ DCs (Fig. 3b).

**LCs are not required to present topical antigen to T cells.** We next examined the contribution of DCs to topical antigen presentation in vivo. To assess the role of migratory versus resident DCs in antigen presentation, we transfered OT-II cells into CCR7$^{-/-}$ or wild-type mice 24 h before topical application of OVA. Proliferation of OT-II cells was minimal in CCR7$^{-/-}$ mice (Supplementary Figure 2), indicating presentation by migratory DCs. We next depleted LCs using anti-CSF1R antibody. Treatment with anti-CSF1R completely depleted LCs from the epidermis (Fig. 4a), although it also reduced other populations, such as macrophages and CD11b+ dermal DCs, as previously

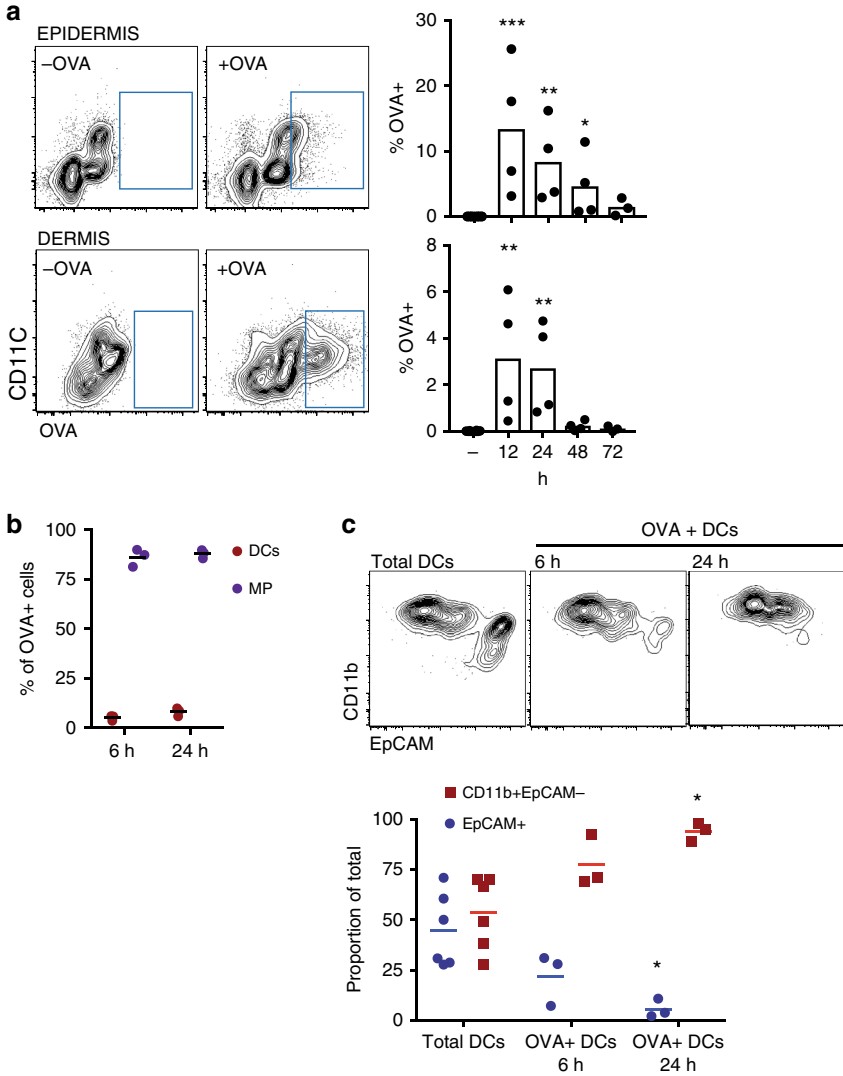

**Fig. 1** Phenotype of CD11c+ MCHII+ cells that are taking up OVA in the skin. **a** Representative plots of OVA+ cells in the epidermis and dermis of Balb/c mice 24 h after applying OVA[AF647] and percentage of OVA+ cells in dermis and epidermis in CD11c+ MHCII+ cells ($N = 3$–4 mice/group). – indicates control mice in which no patch was placed. *$p < 0.05$, **$p < 0.01$, ***$p < 0.001$ vs no antigen. Kruskal-Wallis test with Dunn's multiple comparison. **b** Relative proportion of DCs (red) and macrophages (blue) of all OVA+ cells in the dermis (3 mice/group). **c** Expression of CD11b and EpCAM among total DCs or OVA+ DCs in the dermis. The top shows representative staining, while the bottom shows a summary of at least three mice/group. Circles represent EpCAM+ DCs and squares represent CD11b+ EpCAM−. *$p < 0.05$ vs. frequency in Total DCs. Kruskal–Wallis test with Dunn's multiple comparison

described[23]. Topical OVA was applied to mice treated with anti-CSF1R, isotype control or untreated mice, one day after adoptive transfer of DO11.10 cells. Despite the complete depletion of LCs in the skin, there was no significant impact of anti-CSF1R treatment on antigen presentation to OVA-specific T cells at day 7 (Fig. 4b) after topical OVA. We also examined the impact of anti-CSF1R antibody on antigen capture and presentation by DC subsets purified from draining lymph nodes. Despite the complete depletion of LCs in the skin, there was only a 50% reduction in presentation of antigen by EpCAM+ DCs in the draining lymph node (Supplementary Figure 3), suggesting that this population likely includes langerin+ dermal DCs that are positive for EpCAM, although at lower levels than LCs as previously described[24]. To ablate all langerin+ DCs without altering other DC subsets, we administered diphtheria toxin (DT) to langerin-DTR mice[25] (Fig. 4c). Administration of DT abolishes LCs and langerin-positive dermal DCs, but DCs reconstitute prior to LCs and therefore waiting 7 days after DT administration

results in preferential LC ablation[24]. We administered DT 24 h or 7 days prior to OVA application to abolish LCs and langerin-positive DCs (24 h) or LCs (7 days). With either depletion scenario, and as in the case of anti-CSF1R treatment, there was no significant impact of depletion on the proliferation of OT-II cells in mice lacking LCs (data from 24 h timepoint shown in Fig. 4d). Furthermore, administration of DT 24 h prior to OVA administration had no impact on the development of immune tolerance (Fig. 4e), indicating that langerin+ cells including LCs are not required for Treg generation or primary tolerance to topical antigen.

Our data suggest that LCs are redundant and CD11b+ cDC2s are sufficient for presentation of topical antigen to CD4+ T cells in vivo. To determine if LCs could be sufficient, if not necessary for presentation of topical antigen, bone-marrow chimeras were generated using *MHCII[-/-]* or wild-type donor bone marrow into wild-type recipients (Supplementary Figure 4). Due to radio-resistance of LCs[26], LCs expressed MHCII in both groups, while

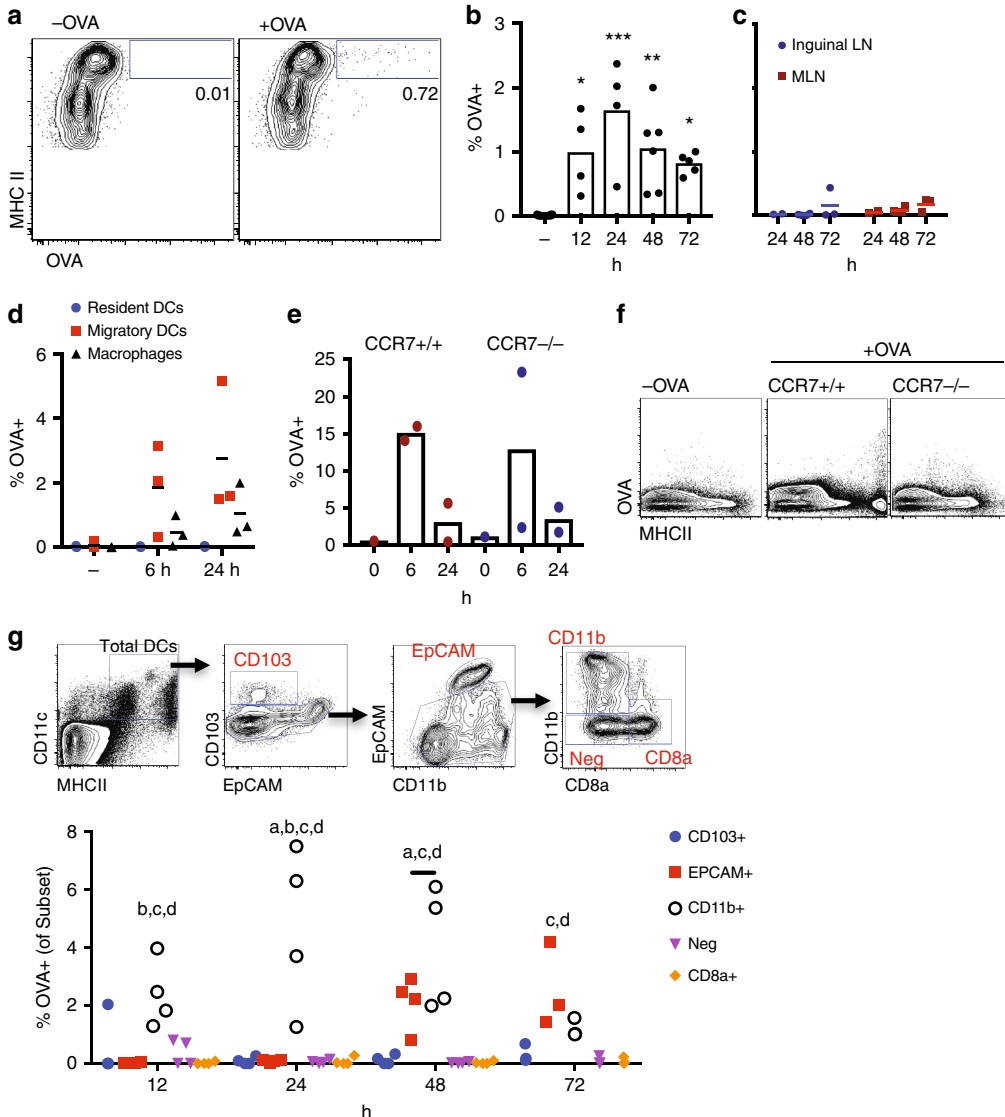

**Fig. 2** Phenotype of DC subsets that are carrying OVA to skin-draining lymph nodes. **a** Gating in CD11c+ MHCII+ cells, representative plots of OVA+ cells in the skin-draining (brachial) lymph nodes from Balb/c mice at 24 h after applying OVA^AF647. **b** Percentage of OVA+ cells in CD11c+ MHCII+ cells from brachial lymph nodes (**b**) and distal inguinal and mesenteric lymph nodes (MLN) (**c**) at different time points. *p < 0.05, **p < 0.01,***p < 0.001 vs. no antigen. Kruskal–Wallis test with Dunn's multiple comparison. **d** Frequency of OVA+ migratory DCs (squares), resident DCs (circles), or macrophages (triangles). **e** Uptake of OVA in dermis of wild-type C57BL/6 or CCR7-/- mice. **f** Representative plots of OVA+ cells in the brachial lymph nodes in wild-type C57BL/6 and CCR7^-/- mice, 48 h after applying OVA^AF647. Representative of three mice per group. **g** Frequency of OVA+ DCs in each DC subset with gating as shown. Letters indicate significant (p < 0.05) difference in frequency as compared to CD103 (**a**), EpCAM (**b**), Neg (**c**), CD8a (**d**) subsets. Two way ANOVA with Tukey's multiple comparisons test

dermal DCs were positive or negative for MHCII. After recipient mice were reconstituted, they received OT-II CD4+ T cells before applying topical OVA. In mice with MHCII expression only on LCs, proliferation of OT-II cells in response to topical OVA was abolished (*WT/MHCII^-/-*), while *WT/WT* chimeras presented high levels of proliferation 3 days after topical OVA application (Supplementary Figure 4). Finally, to determine if CD11b+ cDC2s were necessary for presentation, we used mice lacking IRF4 in the CD11c lineage (*CD11c-Cre x IRF4^fl/fl*) as well as *IRF4^fl/fl* controls. *CD11c-Cre x IRF4^fl/fl* showed a marked absence of CD11b+EpCAM− DCs in the migratory DC compartment and a markedly reduced proliferative response of transferred OVA-specific T cells (Fig. 5a, b). It should be noted that *CD11c-Cre x IRF4^fl/fl* mice have GFP expression in T cells, B cells, and monocytes, as previously described[27], resulting in a partial reduction in IRF4 expression in the

endogenous T cell compartment in addition to the complete reduction of IRF4 in the CD11c compartment. The use of wild-type OT-II cells as responder T cells mitigates the impact of off-target T cell IRF4 depletion, but is a limitation of this model system. Taken together, our results demonstrate that migratory DCs are required for presentation of topical antigen, and indicate a central role for CD11b+ cDC2s, but not LCs in priming of T cells.

**PDL2+ DCs induce LAP+ T cells after antigen exposure**. We studied the phenotype of migratory DCs in the skin-draining lymph nodes after topical antigen delivery using a Viaskin® patch. We did not detect differences in the expression of CD80, CD86, or MHCII, but migratory DCs significantly upregulated expression of PDL2 after topical antigen application (Supplementary Figure 5).

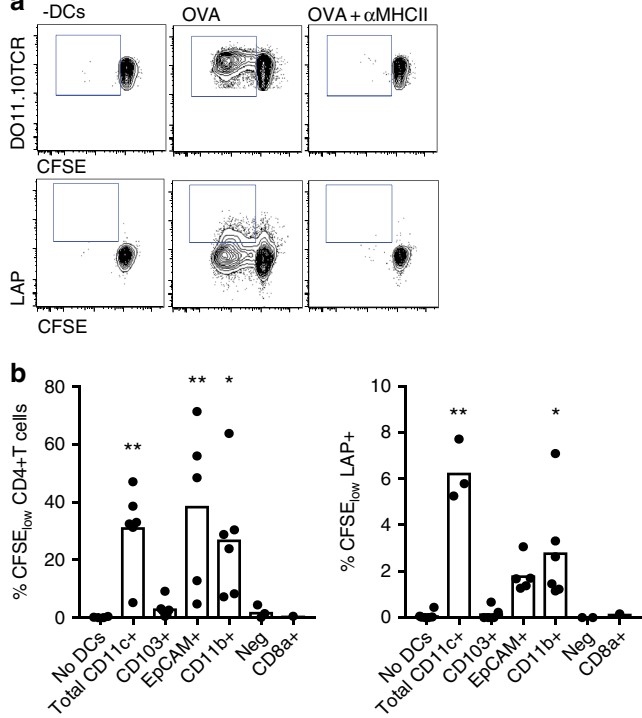

**Fig. 3** Presentation of antigen by skin-draining DC subsets. **a** Proliferation and induction of LAP[+] T cells in the CFSE-low DO11.10 CD4[+] T cells after co-culture with CD11c[+] cells isolated from brachial LN of Balb/c mice receiving topical OVA, in the presence or absence of anti-MHCII antibody. **b** Proliferation and percentage of LAP[+] T cells in the CFSE-low DO11.10 cells after co-culture with sorted DC subsets (ratio 5:1, Tcells:DCs). $N = 4$–5 replicates for populations CD103[+], EpCAM[+] and CD11b[+], 1–2 replicates for populations Neg and CD8a[+]. $*p < 0.05$, $**p < 0.01$ vs. No DCs. Kruskal–Wallis test with Dunn's multiple comparisons. Gating for sorting shown in Fig. 2g

PDL2 and CD301b have been described as markers of skin-draining DCs with Th2 priming potential[12,14]. We observed an increased frequency of PDL2[+] CD301b[−] and PDL2[+] CD301b[+] DCs after topical OVA (Fig. 6a), both in total and migratory DCs, although changes were most evident and statistically significant in the PDL2[+] CD301b[−]population. PDL2 was primarily expressed on migratory DCs, and increased expression was noted on both EpCAM[+] and CD11b[+] DCs after topical antigen, although CD11b[+] DCs had the highest frequency of PDL2 expression (Fig. 6b). We examined the expression of PDL2 and PDL1 on DCs acquiring OVA in the skin and lymph nodes at 6 and 24 h after topical OVA administration. In the epidermis and dermis, we observed a marked up-regulation of PDL2 after patch placement (Fig. 6c), but this was observed in both OVA+ and OVA- DCs indicating that upregulation was not due to the process of antigen uptake. Examination of skin cytokine and chemokine expression by RT-PCR showed a transient increase in inflammatory cytokines and chemokines after patch placement (Supplementary Figure 6), suggesting that this milieu may drive changes in DC phenotype independent of antigen acquisition. When we examined PDL2 expression on migratory DCs in the draining lymph node, we observed a marked elevation of PDL2 only in those carrying antigen (Fig. 6c, d). PDL1 was also increased in OVA + DCs compared to OVA- DCs in the draining lymph node.

We sorted PDL2[+] and PDL2[−] DCs from skin-draining lymph nodes 48 h after topical OVA and co-cultured DCs with DO11.10

CD4[+] T cells. Only PDL2[+] DCs were able to induce proliferation of responder T cells (Fig. 7a), consistent with OVA-carrying DCs being uniformly PDL2[+] (Fig. 6d). Cells proliferating in response to PDL2[+] DCs expressed LAP and CCR4, and low levels of CCR6 (Fig. 7b, c). Only CCR9 was absent from in vitro primed T cells, but otherwise these in vitro primed T cells were consistent with the phenotype of LAP[+] Tregs with multi-tissue homing potential that we previously described as generated in response to topical antigen in vivo[4]. Cytokine production from T cells primed by PDL2[+] DCs was multi-functional, and included both Th1 and Th2 cytokine production (Supplementary Figure 7).

**Dermal DCs acquire topical antigen through the hair follicle**. The lack of requirement for LCs in presentation of topical antigen led us to ask how antigen penetrated to the dermal layers to be acquired by dermal CD11b[+] cDC2s. Hair follicles have been described as an antigen portal, and are densely colonized by skin microbiota that induce Tregs in the skin[28]. We used confocal microscopy to visualize DCs surrounding hair follicles, and extending dendrites into the lumen of the hair follicle (Fig. 8a). Visualization of follicles at steady state and after LC depletion with anti-CSF1R antibody demonstrated that LCs and cDCs were present surrounding the follicle and extended dendrites across the follicle epithelium. To determine if hair follicles were necessary for capture of topical antigen, we used the outbred strain of hairless SKH1 mice, which carry the mutant allele *Hr*, that disrupts the development of normal hair follicles[29]. SKH1 mice were immunocompetent. Examination of the DC subsets in epidermis, dermis and lymph node of SKH1 and Balb/c mice demonstrated a more abundant population of LCs in the epidermis and a greater frequency of DCs in the dermis and draining lymph node of SKH1 mice, but a similar proportion of DC subsets (Supplementary Figure 8). We examined uptake of topical OVA[AF647] in SKH1 or Balb/c mice, and found that antigen delivery to the skin-draining LN was almost completely abolished in SKH1 mice (Fig. 8b). Uptake by dermal DCs was also similarly suppressed in SKH1 hairless mice (Fig. 8b). We next examined whether this lack of antigen delivery in SKH1 mice would impair the generation of tolerance to topical antigen. Indeed, we observed that SKH1 mice treated with the OVA patch did not develop tolerance to OVA, whereas Balb/c mice generated tolerance as shown by suppressed responses to OVA immunization (Fig. 8c). This indicates that the capture of topical antigen resulting in generation of primary immune tolerance is dependent on uptake across the hair follicle niche.

## Discussion

The skin is a barrier site with a resident immune population capable of initiating immunity or tolerance to antigens and microbes. We show that topical application of a model antigen with a Viaskin® patch results in acquisition of antigen by LCs in the epidermis and cDC2s in the dermis, and that these cells transport the antigen to the LN where antigen is presented to naïve T cells and primes LAP[+] Tregs. Viaskin® is an epicutaneous delivery system in which powdered antigen is electrosprayed on the patch surface, and solubilization by natural transepidermal water loss results in efficient delivery to the stratum corneum. It has been extensively tested as a method for allergen desensitization of sensitized mice in preclinical studies[6,30–34] and clinical trials for peanut allergy[35–37], and has also been tested as a method to boost vaccine responses[2,38]. Using this delivery method, we found that antigen uptake by cDC2s in the dermis, that transport antigen to the lymph nodes, leading to generation of immune tolerance was dependent on uptake through the hair follicle. The hair follicle is a unique immune site that contributes to the

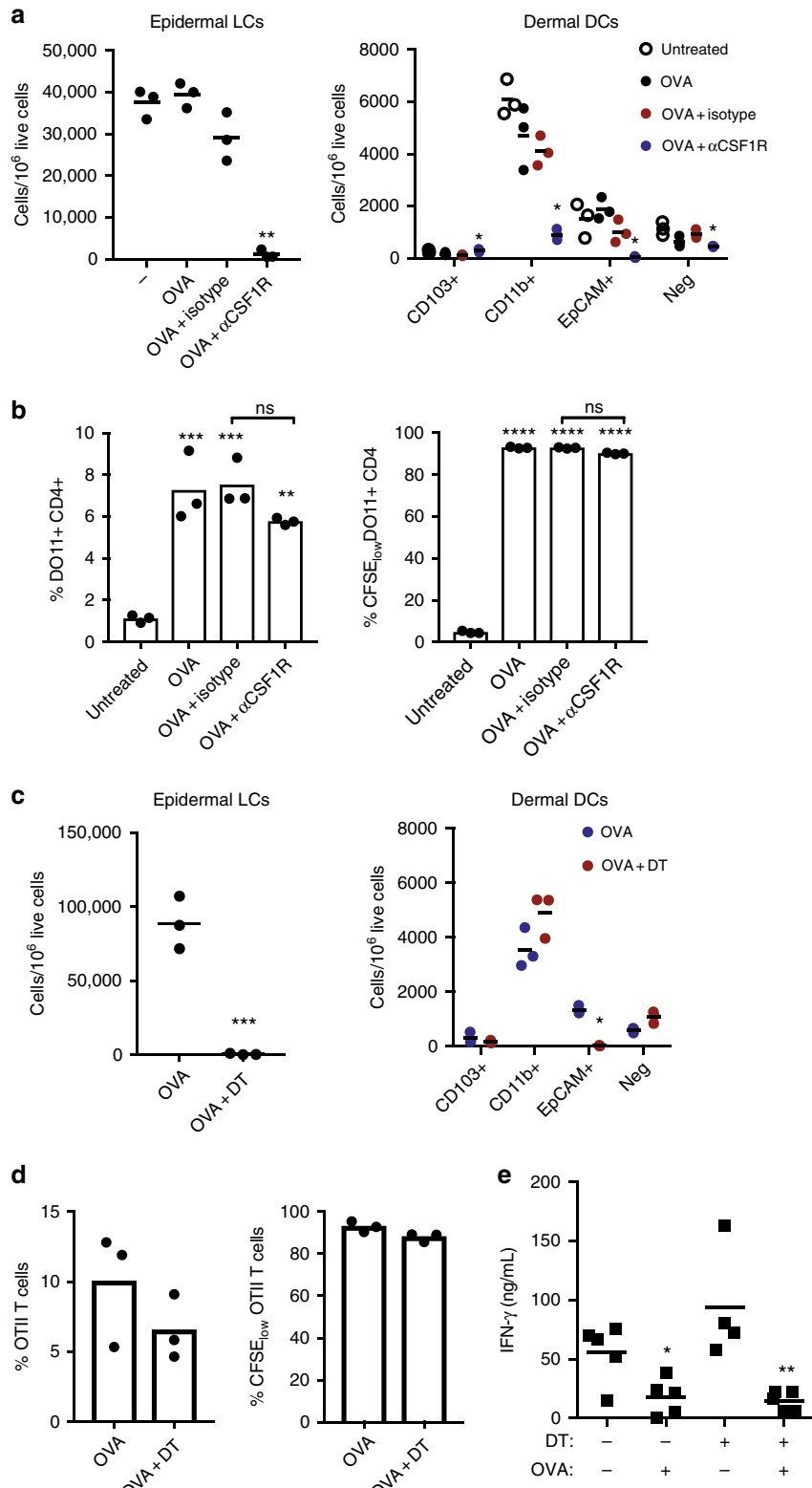

**Fig. 4** Impact of Langerhans cell depletion on antigen presentation. **a** Number of cells per $10^6$ live cells of the different CD11c$^+$ MCHII$^+$ subsets present in epidermis and dermis in the different groups of Balb/c mice. *$p < 0.05$, **$p < 0.01$ vs. isotype control, $t$ test. **b** Percentage of total or CFSE$_{low}$ (proliferated) DO11.10 CD4$^+$ T recovered from brachial lymph nodes 7 days after application of OVA. **$p < 0.01$, ***$p < 0.001$, ****$p < 0.0001$ vs. untreated. ANOVA with Holm-Sidak's multiple comparison test. **c** Number of cells per $10^6$ live cells of the different CD11c$^+$ MCHII$^+$ subsets present in epidermis and dermis in Langerin-DTR mice treated with diphtheria toxin 24 h before applying OVA (OVA + DT) or treated only with (OVA). *$p < 0.05$,***$p < 0.001$ vs OVA, $t$ test. **d** Percentage of total or CFSE$_{low}$ (proliferated) OT-II CD4$^+$ T cells recovered from brachial lymph nodes 7 days after application of OVA. **e** Impact of LC depletion on immune tolerance generated by topical OVA. *$p < 0.05$, **$p < 0.01$ vs. –OVA, $t$ test. $N = 3$ mice/group (**a**–**d**) or 4–5 (**e**)

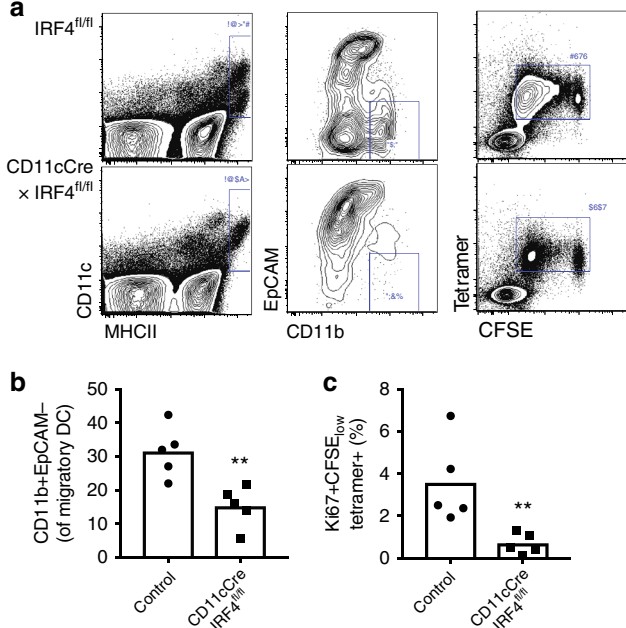

**Fig. 5** IRF4-dependent DCs are required for presentation of topical antigen. Reduction of EpCAM⁻ CD11b⁺ DCs in the lymph nodes of mice lacking IRF4 in the CD11c compartment (representative cytometry **a**, summary **b**). Reduction of T cell priming in mice lacking IRF4 in the CD11c compartment of brachial lymph nodes at Day 3 after topical OVA (representative cytometry **a**, summary **c**). N = 5/group **p < 0.01 vs. control, Mann–Whitney test

regulatory tone of the skin. Shortly after birth, the follicle becomes colonized with a commensal microbiota which drives the recruitment of Tregs into the skin in a CCL20/CCR6-dependent manner[28]. The recruitment of Tregs is necessary for the induction of microbial antigen-specific tolerance[39]. Tregs are also necessary for protection of the stem cell niche in the hair follicle[40]. Chemokine production by follicle epithelial cells is required for the recruitment of Tregs and LCs to the hair follicle[41]. In addition to tolerance, the follicle is also a site of induction of protective immunity. Immunization with naked DNA or nanoparticles induced a robust immune response from the follicle[42,43]. Similar to the Peyer's patch in the intestine, the hair follicle is a unique antigen sampling niche that can be modulated in immune tone by the resident microbiota. In translating these findings to human skin, the density of hair follicles, and potentially the composition of microbial colonization[28,44,45], may be critical factors in determining the efficacy of tolerance induction by topical antigen.

The specialization of skin DC subsets in the generation of tolerance or different forms of immunity has been proposed, and application of antigen with the Viaskin® patch has previously been shown to deliver antigen to LCs[38,46], leading to the hypothesis that LCs mediate the effects of antigen delivered by the epicutaneous route. Targeting of antigen to DC subsets could lead to a more specialized immune response to topical antigen[18,47]. We observed that only two subsets of DCs could acquire and present antigen applied topically, and that one of those subsets was dispensable. This is consistent with previous work that showed delivery to the lymph node by cells that were uniformly CD11b+, but had bimodal expression of CD205[46]. We found that LCs and CD11b⁺ cDC2s could acquire topical antigen, migrate and present to CD4⁺ T cells in the LN, generating LAP⁺ Tregs. cDC1s (CD103⁺CD11b⁻CD8⁺) did not acquire or present

topical antigen to CD4⁺ T cells. We found that LCs were neither necessary nor sufficient to acquire and present antigen to naïve T cells in vivo. In contrast, IRF-4-dependent cDC2s were necessary to present topical antigen to T cells in the draining lymph node. PDL2⁺ CD301b⁺ IRF-4-dependent cDC2s have been described as having strong Th2-priming activity, and poor Th1/Th17 priming activity in response to antigen delivered by the subcutaneous route[12,14]. IRF-4-dependent cDC2s were also identified as critical to the generation of Th2 responses and asthma after application of house dust mite to the skin and challenge through the airways[48]. As described by Murphy and colleagues, IRF-4-dependent cDCs can be further subsetted according to developmental pathways and function[49]. CD11b⁺ cDC2s are dependent on Notch signaling and drive Th17 responses, while CD11b⁻ cDC2s are dependent on Klf4 and drive Th2 responses[50]. Unlike the results reported by Tussiwand et al. that showed significant FITC transport to the LNs by CD11b⁻ cDC2s, we did not observe that this subset acquired labeled OVA, and previous work showed that OVA⁺ DCs in the lymph node were uniformly CD11b positive[46]. However, FITC labels the majority of migratory DCs whereas the fluorescently labeled OVA used in our studies was present in 1.5% of DCs at maximum after topical application. Thus the nature of uptake was not comparable between the studies. Consistent with previous work[12], we observed that PDL2⁺ DCs could elicit Th2 cytokine responses from responder T cells, however we also observed that these DCs could prime LAP⁺ Tregs. LAP⁺ Tregs themselves can secrete IL-4, as we have shown previously[4] and as was first identified as a characteristic of regulatory Th3 cells induced by oral tolerance[20]. Furthermore, our previous studies have demonstrated that the in vivo outcome of topical OVA application to healthy skin of naïve mice using a Viaskin® patch is Treg generation and immune tolerance[4,5], despite the multi-cytokine priming potential of these DCs. Consistent with the regulatory function that we have observed, CD301b⁺ PDL2⁺ DCs have been described to actively suppress the development of T follicular helper cells and the production of antibody responses against protein antigens given without adjuvant[51]. Interestingly, this suppressive activity was highly dependent on the context of antigen presentation. Using the same mice (Mgl2-DTR mice that lose CD301b⁺ DCs after DT injection), no inhibitory effect of CD301b⁺ DCs on antibody production was observed after Nippostrongylus brasiliensis infection[15]. Mice lacking IRF-4-dependent DCs, which overlap with the CD301b⁺ population, show impaired IgE responses to protein antigens (OVA) given with the Th2-priming adjuvant papain[12]. Intranasal immunization has also demonstrated that IRF-4-dependent cDC2s promote the generation of Tfh cells and humoral immunity[52]. The literature clearly supports a key role for IRF-4-dependent cDC2s in allergy and humoral responses, yet a regulatory role is evident under certain conditions. Localization of DCs within the lymph node may play a major role in the outcome of antigen presentation, as has been described by Krishnaswamy et al.[5] Our model of epicutaneous OVA exposure, which can lead to IgE sensitization or immune tolerance[3,4], will be a useful tool for dissecting the mechanisms by which IRF-4-dependent DCs can drive immunologically distinct outcomes.

PDL2 is a marker of cells carrying antigen to the lymph nodes. PDL2 was previously determined to be dispensable for Th2 priming by PDL2⁺ DCs[12] or inhibition of Tfh cells by CD301b⁺PDL2⁺ DCs[51], with the latter being partially dependent on PDL1 and the former dependent on PD1 and OX40 on T cells. We did not explore the mechanism of LAP⁺ Treg generation in this study, but previous literature suggests that PDL2 does not play a major role in T cell priming by these DCs despite being highly expressed. Further work is necessary to elucidate the

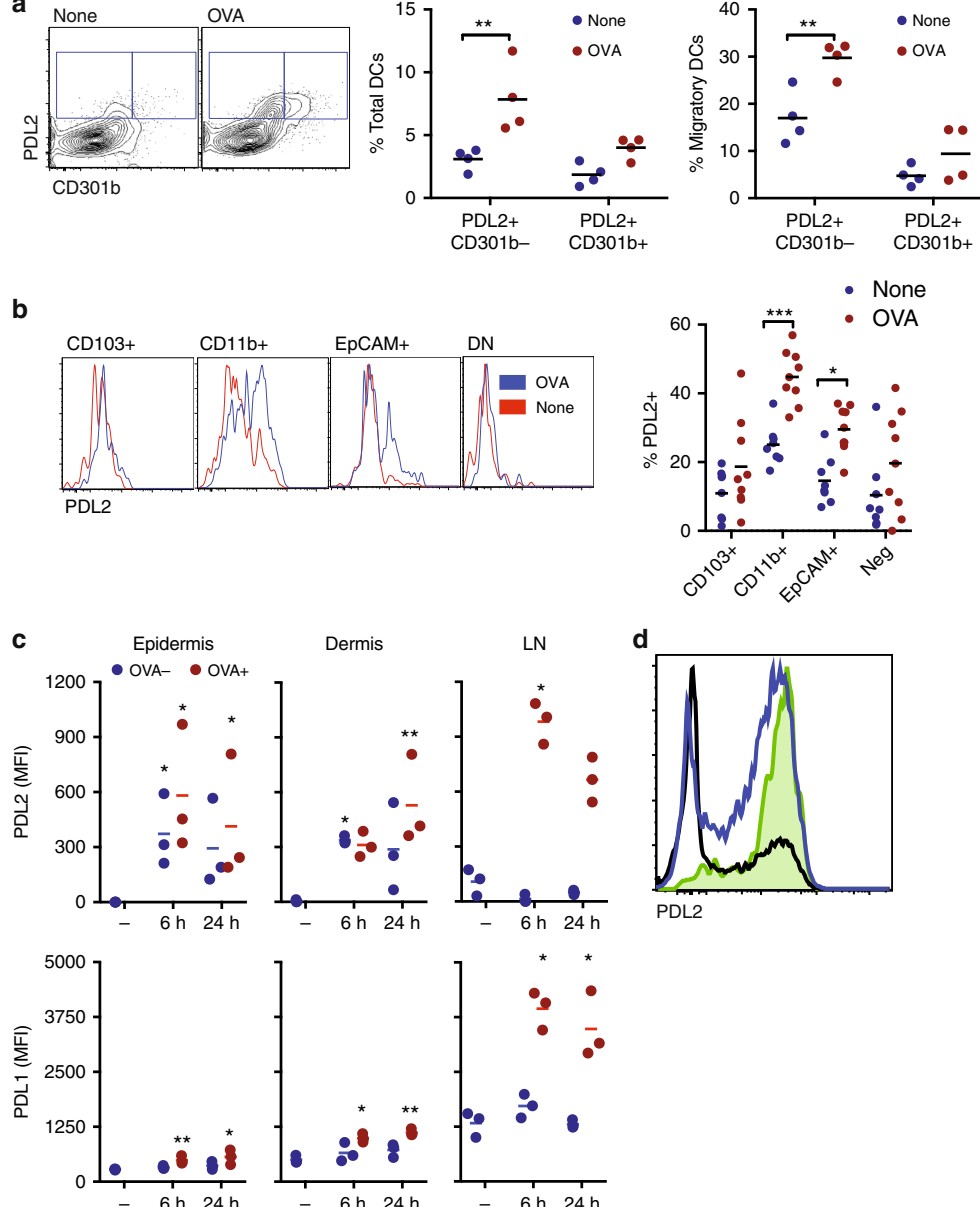

**Fig. 6** PDL2+ DCs in skin-draining lymph nodes after epicutaneous antigen exposure. **a** Expression of PDL2 and CD301b by CD11c+ MHCII+ DCs in brachial lymph nodes of Balb/c mice treated with OVA (red) for 48 h and untreated mice (none)(blue). $N = 4$ mice/group. **b** Representative plots of the expression of PDL2 by the different DC subsets, and percentage of PDL2+ in the different DC subsets. $N = 8$ mice/group. *$p < 0.05$, **$p < 0.01$, ***$p < 0.001$, Two-way Anova, Sidak's multiple comparison test. **c** Expression of PDL2 and PDL1 (median fluorescence intensity, MFI) in OVA+ (red) or OVA−(blue) LCs of the epidermis, or DCs of the dermis or lymph node (LN). Tissues were harvested 6 or 24 h after patch placement, – indicates control mice in which no patch was placed. $N = 3$ mice per group. *$p < 0.05$, **$p < 0.01$, Kruskal–Wallis test with Dunn's comparison. **d** Representative flow cytometry showing PDL2 expression in total DCs (black line), migratory DCs (blue line), or OVA+ DCs (shaded green) in the lymph node

mechanism of LAP+ Treg priming by PDL2+ IRF4-dependent cDC2s.

We observed that LCs captured antigen in the epidermis, and were also present surrounding the hair follicle. They delivered antigen to the LN with delayed kinetics compared to cDC2s, consistent with previously reported slow migration kinetics of LCs[53]. EpCAM+ DCs in the LN likely contained a mixture of LCs and langerin+ dermal DCs and were not fully depleted by anti-CSF1R treatment, but ablation of all langerin-positive DCs (and therefore all EpCAM+ DCs in the LN) in Langerin-DTR mice had no effect on antigen presentation in the draining lymph node, assessed either 3 or 7 days after antigen application. Furthermore,

generation of tolerance was not impaired in the absence of LCs, clearly demonstrating that despite their ability to capture topical antigen, they are not key cells for downstream Treg generation. We have limited our studies to the examination of a single exposure of antigen to healthy skin. In the context of allergic sensitization, or repeated antigen exposures, there may be altered capture and presentation of antigen. Our findings apply to the generation of primary immune tolerance after a single exposure of topical antigen.

In conclusion, we have identified that under homeostatic conditions, topical antigen is acquired through the hair follicle niche and delivered to the LN by a population of PDL2+

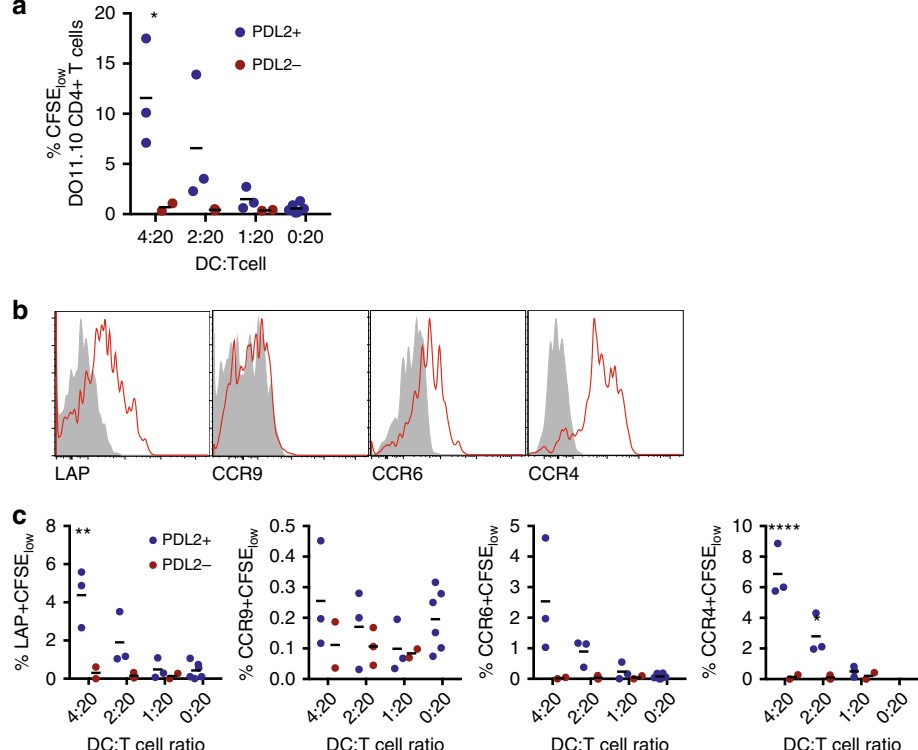

**Fig. 7** PDL2+ DCs from skin-draining lymph nodes present antigen. **a** Percentage of CFSE$_{low}$DO11.10 CD4+ T cells after co-culture with sorted PDL2+ (blue) or PDL2− (red) DCs from mice treated with OVA for 48 h. **b** Representative plots of the expression of LAP, CCR9, CCR6, and CCR4 by CFSE-low DO11.10 CD4+ T cells after co-culture with sorted PDL2+ DCs. Staining control is shown in grey (fluorescence minus one). **c** Percentage of CFSE$_{low}$ LAP+, CCR9+, CCR6+ or CCR4+ CD4+ T cells after co-culture with sorted PDL2+ (blue) or PDL2− (red) DCs from mice treated with OVA patch. $N = 2-6$ replicates. *$p < 0.05$, **$p < 0.01$, ****$p < 0.0001$ vs. PDL2−, Two-way Anova, Sidak's multiple comparison test

CD11b+ DCs to generate LAP+ Tregs with multi-tissue homing potential. The contribution of this unique antigen sampling niche- containing microbes, epithelial cells, DCs and Tregs in close proximity- to the generation of tolerance, allergy, and immunity merits further study.

## Methods

**Mice.** BALB/c and CD45.2+ C57BL/6 mice were obtained from Charles River or The Jackson Laboratory. MHCII$^{-/-}$ and CD45.1+ C57BL/6 were obtained from The Jackson Laboratory. SKH1-Elite mice were obtained from Charles River. CCR7$^{-/-}$, Langerin-DTR mice (originally generated by Kissenpfennig et al.[2], IRF4$^{fl}$ (Stock 009830, Jackson), and CD11c-Cre (Stock 008068, Jackson) were kindly provided by Dr. Miriam Merad and bred at Mount Sinai. DO11.10 mice and OT-II Rag 2 mice were maintained as breeding colonies at Mount Sinai. All procedures were approved by the Icahn School of Medicine at Mount Sinai Institutional Animal Care and Use Committee.

**Patch application.** Mice received a Viaskin® patch (DBV Technologies, Montrouge, France) loaded with 100 µg of OVA, as previously described[4,5]. Briefly, mice were anesthetized, the back was shaved with an electric clipper and then with depilatory cream. 24 h later the patch was placed for 48 h. In some experiments, a patch containing 100 µg of OVA labeled with AlexaFluor647 was applied at different time points. Additional experiments of antigen uptake were performed using OVA labeled with AlexaFluor488 (Invitrogen, Thermo Fisher Scientific, Waltham, MA) applied topically to the skin in solution.

**Primary tolerance.** Mice received a Viaskin patch loaded with 100 µg of OVA as above for 48 h, or as control the skin was prepared in the same manner but no patch applied. Seven days after patch application, mice were injected in each hock with 50 µg of OVA in Complete Freunds Adjuvant (Sigma Aldrich, St. Louis, MO), and after another 14 days boosted in the hock with 50 µg OVA in incomplete Freund's Adjuvant (Sigma Aldrich). After an additional 7 days, the popliteal lymph nodes were harvested and $1 \times 10^6$ cells were re-stimulated with 50 µg/ml of OVA for 72 h. Supernatants were harvested and IFN-γ measured by ELISA (eBioscience, Thermo Fisher Scientific, San Diego, CA).

**Adoptive cell transfer experiments.** CD4+ T cells from DO11.10 or OT-II mice were purified by negative selection (StemCell Technologies, Vancouver, British Columbia, Canada), labeled with CFSE (Invitrogen, Thermo Fisher Scientific, Waltham, MA) and $3 \times 10^6$ CD4+ T cells were transferred i.v. into recipient mice. 24 h later, mice were exposed to OVA by OVA-patch for 48 h. Three days or 1 week after first exposure, brachial, inguinal and mesenteric lymph nodes were harvested for assessment by flow cytometry.

**Antigen presentation assay.** Dendritic cell subsets were sorted or total CD11c+ cells were purified by using CD11c microbeads (Miltenyi Biotec, San Diego, CA) 48 h after applying the OVA patch. DCs were cultured at a ratio of 1:5 with DO11.10 CD4+ T cells labeled with CFSE. After 3 days, T cell proliferation, LAP+ T cell induction and tissue-homing markers expression were assessed by flow cytometry. In some experiments, blocking anti-MHCII antibody (M5/114.15.2) (eBioscience, Thermo Fisher Scientific) was added to the well at 10 µg/mL.

For cytokine quantification in the supernatant, after 4 days of co-culture, cells were re-stimulated with anti-CD3/CD28 (eBioscience, Thermo Fisher Scientific) for 3 days. Supernatants were harvested and cytokines measured by ELISA according to manufacturer's instructions (all from eBioscience).

**Purification cells from dermis/epidermis.** Skin from the back or from the ears was incubated with 2.5 mg/mL of dispase II (Roche) in PBS, at 4 °C overnight, with dermal side facing down. After washing with PBS, dermis and epidermis were separated and digested separately with 1 mg/mL of collagenase D (Roche) and 1 mg/mL DNase I (Roche) for 1 h at 37 °C[54]. After digestion, samples were homogenized with a 18G needle and filtered.

**OVA tetramer staining.** In experiments utilizing OT-II cells, OVA-specific CD4+ T cells were stained by using I-A(b) OVA$_{323−337}$ tetramer labeled with PE, kindly provided by the NIH Tetramer Core Facility. After harvesting the cells, they were stained with 6 µg/mL of the tetramer for 30 min at 37 °C, before performing surface staining.

**Flow cytometry.** Lymph nodes were isolated and stained with specific antibodies according to standard techniques. For dendritic cell isolation, lymph

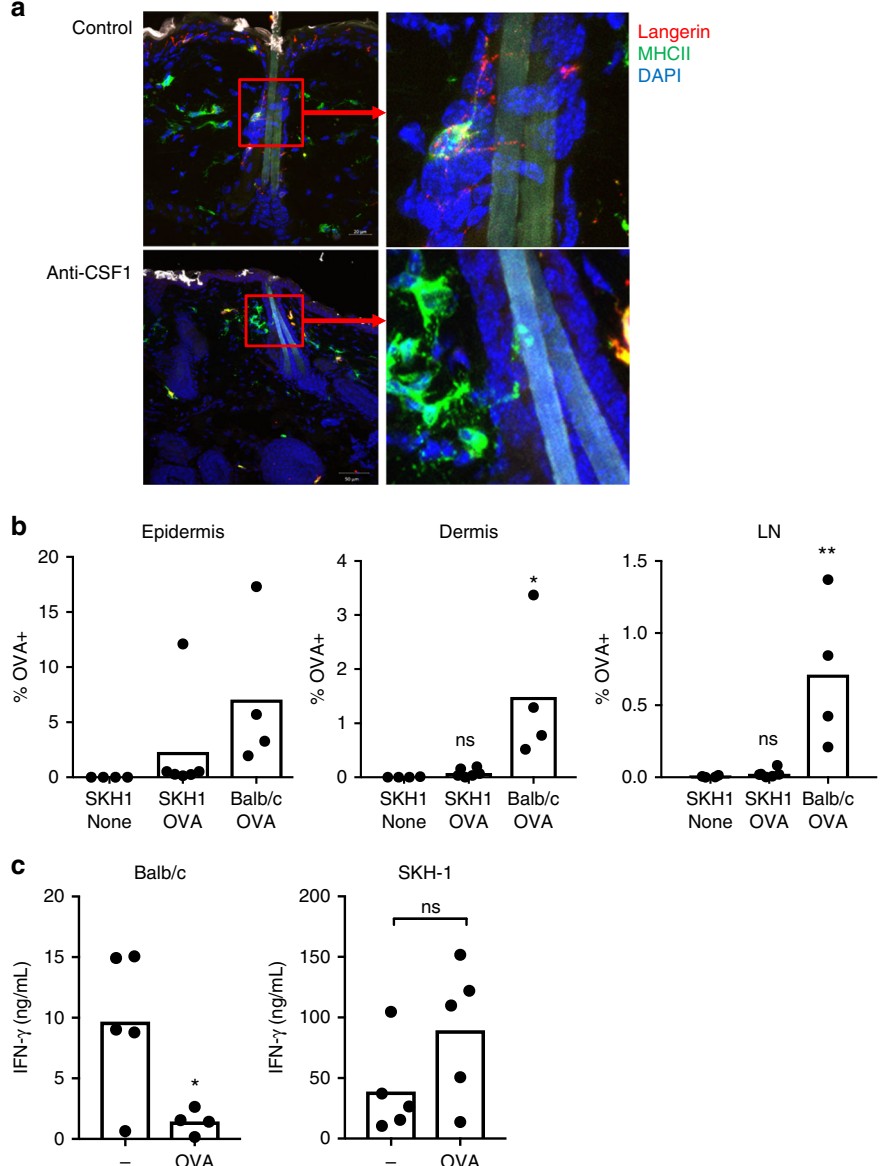

**Fig. 8** Dermal DCs acquire topical antigen through hair follicles. **a** Confocal microscopy of skin 24 h after patch application in a wild-type Balb/c mouse and mouse treated with anti-CSF1R antibody to deplete Langerhans cells. Scale bars, 20 μm in control and 50 μm in anti-CSF1R. **b** Percentage of OVA+ cells in CD11c+ MHCII+ cells in the epidermis, dermis and lymph nodes (LN) 24 h after applying OVA[AF647]. $N = 4$–6 mice/group. $*p < 0.05$, $**p < 0.01$, ns, non-significant, One-way Anova, Tukey's multiple comparison test. **c** Immune tolerance induced by topical OVA in SKH-1 and Balb/c mice ($N = 4$–5/group). $*p < 0.05$, ns, non-significant, $t$ test

nodes were digested with 0.5 mg/mL of collagenase D (Roche) and 0.1 mg/mL DNase I (Roche) for 25 min at 37 °C. Ten micrometer EDTA (Invitrogen) was added for the last 5 min. All antibodies were from Biolegend (San Diego, CA) unless otherwise stated. Cells were blocked with FcBlock before staining with surface antibodies and live/dead Fixable Blue (Invitrogen). For intracellular staining, cells were fixed and permeabilized, using Foxp3-optimized buffers from eBioscience. For Treg staining, the antibodies used were recognizing CD3 (17A2, cat. 100222 and 100232), CD4 (RM4–5, cat. 100526 and 100531), LAP (TW7–16B4, eBioscience cat. 46–9821–82), CD25 (PC61.5, cat. 102024), Foxp3 (FJK-16s, eBioscience cat. 48–5773–82) and DO11.10 TCR (KJ1–26, eBioscience cat. 17–5808–80). For tissue-homing marker staining, we used antibodies recognizing CCR9 (CW-1.2, cat. 128712), CCR6 (29–2L17, cat. 129819) and CCR4 (2G12, cat. 131204). Dendritic cell subsets and macrophages were differentiated based on staining with antibodies against CD45 (30-F11, cat. 103128 and 103116), CD11c (N418, cat. 117328 and 117335), MHCII (M5/114.15.2, cat. 107622 and 107620), CD103 (2E7, cat. 121406 and 121420), EpCAM (G8.8, cat. 118216), CD11b (M1/70, cat. 101257), CD8a (53–6.7, eBioscience cat. 56–0081–82), CD24 (MI/69, cat. 101808), F4/80 (BM8, cat. 123127), CD64 (X54–5/7.1, cat. 139316) (all from Biolegend), and Langerin

(caa8–28H10, Miltenyi Biotec cat. 130–102–169). Dendritic cell phenotype was studied using antibodies against CD45.1 (A20, cat. 110714), PDL2 (122, eBioscience cat. 11–9972–82) and PDL1 (MIH5, eBioscience cat. 46–5982–82), CD301b (URA-1, cat. 146814), CD80 (16–10A1, cat. 104732) and CD86 (GL-1, cat. 10503). All antibodies were used at a concentration following manufacturer's recommendations. Cells were acquired on a LSR Fortessa cytometer (BD Bioscience, San Jose, CA) and analysis performed using FlowJo software (Ashland, OR).

Sorting of skin-draining lymph node DC subsets was performed in the CD11c+ MHCII+ population after gating out T and B cells with CD3/CD19 staining. Cells were FACS sorted on a CSM4L (BD Bioscience).

**Depletion of Langherhans cells**. Naïve BALB/c mice were pre-treated with anti-CSF-1R antibody (AFS98) (2 mg) or isotype control (both from BioXCell, West Lebanon, NH) intraperitoneally 8 days before applying the OVA patch. 500 μg of antibody were administered every 3 days until the end of the experiment.

Langerin-DTR mice were injected i.p. with 1 μg of diphtheria toxin (Sigma-Aldrich) to deplete Langerin+ cells, as previously described[25,55].

**Bone marrow chimeras**. Bone marrow chimeras were performed by irradiating 8−week old CD45.1$^+$ C57BL/6 recipient mice with 2 doses of 600 rad separated by 16 h. After 4 h, bone marrow cells (>5 × 10$^6$ cells) isolated from CD45.2$^+$ WT or MHCII$^{-/-}$ mice were injected i.v. Chimerism was analyzed in blood after 6 weeks by looking at the percentage of CD45.2$^+$ cells versus CD45.1$^+$.

**Microscopy preparations**. Skin from the back of the mice was harvested and frozen directly in OCT. 8 µm sections were taken, fixed in 4% paraformaldehyde (Sigma-Aldrich) for 10 min at RT and permeabilized with 0.1% TritonX-100 for 15 min. Skin sections were blocked before staining with 2.5% Normal Goat Serum (Sigma-Aldrich) and 1% BSA for 1 h at RT. Staining was performed with anti-MHCII (1:200 dilution, clone 2G9, BD Pharmingen, San Diego, CA, cat. 553623) and anti-Langerin (1:50 dilution, clone eBioL31, eBioscience cat. 12–2075–82) overnight at 4 °C. Sections were mounted using Pro-Long™ Gold antifade reagent with DAPI (Invitrogen). Images were obtained with a Zeiss LSM780 confocal microscope and analyzed with ImageJ and ZEN (Zeiss) software.

**Statistics**. Differences between groups were analyzed by Mann–Whitney $U$-test, one-way or two-way ANOVA test followed by post-hoc analysis with the Sidak's or Tukey's multiple comparisons test. Data analysis was done by using Prism software (GraphPad, San Diego, CA). Results are expressed as mean ± SEM. A value of $p < 0.05$ was considered significant.

## Data availability

The source data for Figs. 1a–c, 2b–e,g, 3b, 4a–e, 5b–c, 6a–c, 7a, c, 8b, c are provided in Source Data files in the Supplementary material. Microscopy images and fcs files are available from the communicating investigator upon request. A Reporting Summary for this Article is available as a Supplementary Information file.

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

## Acknowledgements

We thank the Microscopy Core at Icahn School of Medicine at Mount Sinai, in particular Dr. Nikos Tzavaras, for assistance with confocal experiments and the Flow Cytometry Core for assistance with cell sorting. We also thank the NIH Tetramer Core Facility for providing the I-A(b) OVA$_{323\text{-}337}$ tetramer. Financial support for the study was provided by NIH grants AI093577, AI124062 (to M.C.B.), K08DK102978 (to D.D.), the Robin Chemers Postdoctoral Fellowship (to L.T.), and a postdoctoral fellowship from the Spanish Fundación Alfonso Martín Escudero (to D.L.O.).

## Author contributions

L.T. and D.L.O. designed, performed experiments and analysed the data. D.D. performed experiments. D.D., L.M., J.A., M.M., and H.A.S. provided ideas and comments about the manuscript. L.T. and M.C.B wrote the manuscript. M.C.B. conceived the idea and supervised the project.

## Additional information

**Competing interests:** L.M. and H.A.S. are employed by DBV Technologies, the maker of the Viaskin Patch used to deliver topical antigen in this manuscript. D.D. has received research support from DBV Technologies. The remaining authors declare no competing interests.

