## [Peer Review File · Nature Communications]

Reviewers' comments:

Reviewer #1 (Remarks to the Author):

Summary Tordesillas et al.

The presented manuscript by Tordesillas et al. presents an interesting mechanisms of skin resident Treg induction dependent on dermal cDC2s and their interaction with an intact hair follicle. However interesting the manuscript has certain short comings which should be addressed prior to publication.

Major points:

Figure 1

A A comprehensive gating strategy in conjunction with the uptake of OVA is missing. As the myeloid cell compartment of the dermis is heterogeneous (Tamoutounour et al. Immunity, 2013) this should be taken into account and the OVA uptake of all myeloid cell subsets resident to the dermis should be assessed.

C The presented flow cytometry is of poor quality and should be refined with proper voltage settings and clearer gating strategies.

Figure 2

A Again here it is not clear how the flow cytometry gating was done. The authors should include proper gating strategies wherever possible. Is this data specifically gated on migratory DCs? Furthermore, please include an analysis of the OVA uptake in the other lymph node resident myeloid fraction such as the resident DCs and the lymph node resident macrophage.

D Have the authors analysed the dermal uptake of OVA in CCR7ko mice. Is the dermal myeloid cell compartment of CCR7ko mice comparable to a WT compartment?

Figure 3

A As both Epcam⁺ and CD11b⁺ DCs induce DOI11.10 T-cell proliferation, as their any qualitative difference in the cytokine responses of the cells primed by either Epcam⁺ or CD11b⁺?

A To me it looks from the data presented that the LAP⁺ T-cells are induced during the later proliferation cycles of the T-cells. Could the authors provide a more detailed analyses of the proliferation cycle in conjunction with the expression of LAP, Foxp3 and Helios? Also compare this to the induction of LAP⁺ T-cells by Epcam⁺ cells.

Figure6

Cytokine profile of PDL2⁻ vs PDL2⁺ cDC2 what are determinants for Treg induction, either protein or RNA seq would be nice to inform the mechanisms and differences between these two DC phenotypes

What is the kinetic of PDL2 and CD301b induction on cdc2 after OVA?

What happens when PDL2 or CD301b is blocked during OVA uptake? Is Treg generation changed?

Are effector T-cells generated?

Are the generated Tregs suppressive ex vivo?

Figure 8

Please show the characterization of the skin myeloid compartment of the SKH1-elite mouse as this is an outbred strain.

How does the distribution of DCs look in the skin of hairless mice? Is clustering of DCs visible after OVA application? Or is this orchestrated by a hair follicle dependent process.

Reviewer #2 (Remarks to the Author):

The authors report an interesting site by which CD11b⁺PDL2⁺ cDC2 acquire antigen delivered by patch, via the skin hair follicle, to induce LAP⁺ regulatory T cells. Given the context of their previous work this has important implications for developing systems to induce food tolerance and therefore could be of interest to a broad scientific community.

They provide a thorough characterization of the nature of the DC subsets that acquire and present antigen from the Viaskin patch system. The first two figures establish that the DC subsets that acquire and carry protein antigens from the skin to draining LNs behave in a similar manner to what has been previously shown for other forms of immunization through the skin (Levin et al J. Invest. Derm 2017 v.137). Further, the nature of the DCs that induce CD4⁺ T cell activation (CD11b⁺ cDC2 and Epcam⁺ LC) are consistent with prior results in other sites (Dudziak and Nussenzweig et al Science 2007). However, this is a unique system and warrants the characterization performed and the authors execute clean, well-controlled experiments.

The conclusion about the role of LCs in T effector responses has been contentious and potentially depends on the nature of the immunization and the systems used to manipulate LCs. Tordesillas et al demonstrate that LC are indeed CD4⁺ T cell stimulatory, but like cDC2s, also induce a significant LAP⁺ regulatory population (Fig. 3). But they are not necessary or sufficient for such responses when deleted using anti-CSF1R and Langerin-DTR (Fig. 4) or conversely when MHCII is removed from non-LC.

Although their previous work has established the role of the LAP⁺ Tregs in this patch model, they do not show functionality of the LAP⁺ T cells induced in vitro by the implicated DCs. Indeed, using their in vitro T cell activation approach they demonstrate Th1, 2 and 17 cytokines (despite referring in the text to a mixed Th1/Th2 response) along with LAP expression, therefore it is unclear what the final in vivo T cell response would actually be. Adoptive transfer experiments might address such questions.

Despite the rarity of these cells, they go on to elegantly show, using flow cytometry and immunofluorescence, that the relevant DC population, CD11b⁺ cDC2s, express PDL2 and access epicutaneously applied antigen via the hair follicle. This novel finding suggests how induction of a regulatory rather than inflammatory setting might be established with patching in the absence of adjuvant or skin disruption. Finally, the use of hairless SKH1 mice presents a new approach for studying the function of skin DCs. As such, the characterization of DCs in these mice is warranted (data not shown is indicated in the manuscript). The authors should test the logical implication of the findings presented in Fig. 8: is there a functional difference in the immune response to patched antigens in these mice?

Minor comments:

1. The paper is under-referenced. Perhaps this is due to journal restrictions, but certain previous findings are not adequately referred to and/or findings are not put into context of previous work. In addition to some of the references above, the complicated models for LC vs. langerin⁺ dermal DC depletion using different DTR mice is not referenced (Yao et al 2015 JACI v136 p1387) or discussed. Also, the conflicting data on the role of cDC2s in Tfh priming is not adequately addressed in the discussion (Kumamoto et al, ref 7 vs. Krishnaswamy et al Sci. Immunol 2017). The role of LAP⁺ Tregs is not discussed outside of a conclusion from their previous work (Tordesillas et al 2016, ref 12).
2. Figs. 1-4 and most of the supplementary material do not show statistics.
3. The Viaskin patch system is not described and therefore the background on why such a system might promote Tregs is not clear.
4. In the discussion, the authors reference PDL2⁺ Cd11b⁺ DCs driving Th2 but not IgE responses,

yet Gao et al (ref 5) demonstrated impaired IgE in IRF4^{-/-} mice.

Reviewer #3 (Remarks to the Author):

Epicutaneous immunization, i.e., the topical application of an antigen / vaccine onto the skin is a desired option in vaccinology. In spite of many years of research, no generally accepted procedure, that would replace standard injections on a widespread (global) scale, has emerged.

Here, the authors investigate antigen uptake through the skin using a VIASKIN patch that delivers antigen topically. Their goal is to develop treatments for allergy based on that very epicutaneous application of an allergen, hypothesizing that the epicutaneous route favors induction of regulatory T cells. So, the questions addressed are timely and they are important and relevant. The authors are studying the antigen uptake and the ensuing T cell immune responses through the VIASKIN patch in a MOUSE MODEL.

In extension of a number of preceding papers the novel findings in this manuscript are:

- the detailed characterization of those skin DCs that take up the model antigen OVA (ovalbumin)
 - it is Langerhans cells and CD11b⁺ dermal dendritic cells, but not langerin⁺/CD103⁺ cDC1 nor lymph node-resident CD8⁺ DCs;
- the observation that the Treg-inducing skin dendritic cells are PDL2 and CD301b positive.
- The observation that LCs, although being able to take up antigen, do not contribute to antigen-specific T cell proliferation nor to development of Tregs.
- and the demonstration that hair follicles are crucial (experiments with hairless mice).

I HAVE SOME COMMENTS, QUESTIONS AND SUGGESTIONS:

1. The authors show uptake of OVA by LCs and they show that OVA can be detected in the draining lymph nodes. Probably not unexpectedly, the percentage of LCs that takes up OVA in situ is quite low (between 6 and 16% of all LCs). So are the corresponding percentages of OVA-carrying DCs in the draining LNs. Some points regarding these observations may need a little more documentation or reporting of observations / data:

1A. The Viaskin patch does not contain an adjuvant. What is the migration stimulus for the LCs (and dermal DCs)? Is the occlusive milieu under the "lid" of the patch inflammatory? LPS-containing OVA? A simple H&E histology may give some clues if there is inflammation - influx of lymphocytes and / or neutrophils.

1B. For the sake of completeness, the authors should report whether epidermal keratinocytes also took up some OVA. In the present figure they were gated away. (I realize, that this information is contained in the authors' previous JI paper from 2011 - Dioszeghy et al. This paper is not cited here, though. Please cite it.)

1C. Connected to the previous point, it would be important to know whether LCs indeed emigrate from the epidermis in substantial, measurable numbers. This could be easily tested by either looking at MHCII- or Langerin-stained epidermal sheets or by analyzing epidermal cell suspensions for their percentages of MHCII or langerin⁺ cells. Sheets may be more straightforward. If emigration is observed, how would it compare to what is published for explant cultures (where the

vast majority of LCs leave the epidermis over a period of about 3 days) or for CHS experiments, where the numerical reduction of LCs can also be readily observed and quantified. (Again, I realize that this was looked at in the not cited previous JI paper from 2011 - Dioszeghy et al. - Fig.4C. Please mention.)

1D. The authors show that OVA+ cells in the LNs reside in the MHC-high/CD11c fraction. How are the percentages of OVA+ cells in this fraction? In Fig.2A the authors look at total DCs where percentages are low. In the migratory fraction the percentages should be markedly higher. This information should be retrieved from the available data and presented.

1E. Langerhans cells in the lymph nodes. In Fig.2E it becomes apparent that a population of EpCAM+ cells, i.e. LCs carry OVA. What about the inverse? Do all LCs in the node carry OVA? Probably not, since even in the epidermis only a small part of LCs take up the OVA. Is it then correct to read Figure 2F such that of 20,000 EpCam+ cells only about 50 carry OVA 48hrs after application? This would mean only 0.25%? Correct? If so, please state in an unequivocal and affirmative way. If not, please try to word it such that misunderstandings will be avoided. I think it is important to report such numerical relations / proportions.

2. Referring to data from Figure 6 - PDL2 and CD301b. These markers are expressed at higher levels in the migratory fraction of skin draining nodes (Fig.6B). Since this paper emphasizes the epicutaneous aspect (and also because Gao-Medzhitov in ref.7 show that PDL2/CD301b+ cells contain LCs and dermal DCs / and not least also because DVB Technologies emphasizes Langerhans cells in their website description) it would be interesting and relevant to explore this point more in detail. Two suggestions: (A) In Fig.1 the authors show some phenotype of skin cells after 48 hrs under the VIASKIN patch - both epidermal LCs and dermal DCs. Do these DCs turn PDL2 and/or CD301b positive? Both fractions - the OVA-positive and the OVA-negative fraction? (B) This may be technically not possible? Place the VIASKIN patch onto an explant culture and look at "crawl-out" DCs. In this context: A previous paper by Pieter Leenen's group has shown up-regulation of CD301 on LCs upon explant culture. Back then, CD301a versus CD301b could not be distinguished, though (J.Leukocyt.Biol. 80:839, 2006).

3. Referring to data from Figure 6 and 7 - combined. In 6A the upregulation of PDL2 is shown. In 7A the proliferation of DO11 cells induced by sorted DCs from the draining nodes is depicted. PDL2-negative cells do not induce proliferation. Why? Do they not take up the OVA? This could be directly assessed using the Alexa647-OVA patch.

4. Significance asterisks are only from Figure 5 onwards. Does this mean that none of the important data on antigen uptake and carriage in Figure 1 - 4 are significant? Please clarify in an unequivocal way for each figure.

5. How many times was the experiment in Suppl. Fig 1 (CCR7-/- mice) done?

6. Suppl. Figure 2. Please clarify: Is "B" the summary of all 4 experiments, one representative of which is depicted in "A"? The legend is somewhat confusing in this regard. This must be stated because %CFSE of CD11+ and EpCAM+ cells in "A" appear quite similar, in "B", however, they look very different.

7. Figure 8A. Beautiful pictures - good resolution! Yet, not sure how to interpret them. I cannot really see the grey for the OVA (A647-OVA, I assume). Or is it the light blueish in the high magnifications? If so, a few little arrows may help. Moreover, in the anti-CSFR1-treated mice langerin is absent but there are grey DCs in the epidermis? This is confusing, since LCs are / should be depleted.

8. The authors may want to briefly touch upon the "change of paradigm" between their 2011 JI paper, where LCs still were the suspects (although without experiments directly showing this) and

this manuscript, that rules out LCs as mediators of the clinically observed effects of EPIT.

MINOR POINTS

9. M&M. Please specify which langerin-DTR mice were used. The ones generated by Kissenpfennig/Malissen or those by Clausen/Bennett or were they made "in house"?

10. The authors write: "In vitro loading of all DC subsets with OVA peptide as positive control led to proliferation (not shown)." This important control should be shown in a supplementary figure.

11. Figure 2E. Perhaps the authors may want to give some words of explanation for the gate in the upper left panel that looks a little strange at first glance. It was chosen that way in order to include both migratory LNDCs (MHCII-high) and resident LNDCs (MHCII-lower). Correct?

12. Fig.4. What is the difference between panels "C" (% CFSE-low DO11 cells) and "D" (% OVA-specific CD4 cells). CFSE seems clear to me. But what about "D"? Is the legend wrong in that "D" would be the same as "F", i.e., pentamer-positive endogenous CD4 cells? This would make sense. Please get this right.

13. The authors write: "Despite the complete depletion of LCs in the skin, there was only a 50% reduction in presentation of antigen by EpCAM+ DCs in the draining lymph node (Supp.Fig 2), suggesting that this population could also include langerin+ dermal DCs that upregulate EpCAM after migration to the LN." Is there any published evidence that the dermal cDC1s (CD103+/langerin+) upregulate EpCAM? If so, please cite. If not please state that this is a speculation only. Either way is fine.

14. Suppl.Fig.3 - FACS analyses. Please show a representative dot plot for the PD-L2 staining.

15. Legend to Fig.7B. By "staining control" the authors mean "isotype control", I guess. Correct? Please state explicitly.

16. Adding the word "outbred" to the SHK1 Elite hairless mice in M&M might preempt the question for functional T cell assays in these mice.

We would like to thank the reviewers for their positive and constructive comments that we have used to revise and improve our manuscript. We have added several new pieces of data that strengthen the findings, including performing skin tolerance experiments in Langerin-DTR and SKH-1 hairless mice, and testing the requirement for IRF4-dependent cDC2s in presentation to T cells. In addition, we have examined the inflammatory response of skin to patch placement, and performed additional analysis of SKH-1 skin. We have also generated additional data to analyze PDL1 and PDL2 expression of cells that acquire antigen in the skin and lymph node. We believe these additional data strengthen our conclusion for the central role of the hair follicle and IRF4-dependent PDL2+ cDC2s in the induction of tolerance to topically administered antigens. Point by point responses to all reviewer comments are provided below.

Response to Reviewers

REVIEWER 1

Summary Tordesillas et al.

The presented manuscript by Tordesillas et al. presents an interesting mechanism of skin resident Treg induction dependent on dermal cDC2s and their interaction with an intact hair follicle. However interesting the manuscript has certain shortcomings which should be addressed prior to publication.

Major points:

Figure 1

A A comprehensive gating strategy in conjunction with the uptake of OVA is missing. As the myeloid cell compartment of the dermis is heterogeneous (Tamoutounour et al. *Immunity*, 2013) this should be taken into account and the OVA uptake of all myeloid cell subsets resident to the dermis should be assessed.

Response: *We have now included a gating strategy in the Supplemental Figure 1 that demonstrates gating for epidermal LCs, dermal DCs and macrophages, and lymph node DCs and macrophages. Figure 1B (new) now quantifies uptake in dermal DCs and macrophages.*

C The presented flow cytometry is of poor quality and should be refined with proper voltage settings and clearer gating strategies.

Response: *This has been replaced.*

Figure 2

A Again here it is not clear how the flow cytometry gating was done. The authors should include proper gating strategies wherever possible. Is this data specifically gated on migratory DCs? Furthermore, please include an analysis of the OVA uptake in the other lymph node resident myeloid fraction such as the resident DCs and the lymph node resident macrophage.

Response: *This was gated on all DCs to determine if uptake was also occurring in resident DCs. A gating strategy is now shown in Figure S1. Figure 2D (new) shows uptake by migratory DCs, resident DCs, and CD64+ macrophages.*

D Have the authors analysed the dermal uptake of OVA in CCR7ko mice. Is the dermal myeloid cell compartment of CCR7ko mice comparable to a WT compartment?

Response: *Figure 2E (new) now summarizes dermal uptake by DCs in CCR7 +/+ and -/- mice. In data not shown, we have verified that the distribution between dermal DCs and macrophages is similar in CCR7+/+ and -/- mice.*

Figure 3

A As both Epcam+ and CD11b+ DCs induce DOI11.10 T-cell proliferation, is there any qualitative difference in the cytokine responses of the cells primed by either Epcam+ or CD11b+?

Response: *Because we did not find any evidence of an in vivo role of antigen presentation by LCs in this context (MHC II expression only on LCs did not allow T cell priming, and as we now show in the manuscript, depletion of LCs did not affect development of tolerance) we did not focus any further on the outcome of T cell priming by EpCAM+ cells in vitro.*

A To me it looks from the data presented that the LAP+ T-cells are induced during the later proliferation cycles of the T-cells. Could the authors provide a more detailed analysis of the proliferation cycle in conjunction with the expression of LAP, Foxp3 and Helios? Also compare this to the induction of LAP+ T-cells by Epcam+ cells.

Response: We agree that the mechanism of LAP+ T cell induction is interesting but not explored in this manuscript. In order to keep the focus of the manuscript from becoming too diffuse, we have not addressed the mechanism of LAP+ Treg induction and instead have focused on the mechanisms of antigen presentation. Because we find no role of LCs in LAP+ Treg induction in vivo, we have not pursued additional mechanistic in vitro experiments with this subset.

Figure 6

Cytokine profile of PDL2- vs PDL2+ cDC2 what are determinants for Treg induction, either protein or RNA seq would be nice to inform the mechanisms and differences between these two DC phenotypes

Response: We agree. We have done exploratory analysis by PCR to examine differences in gene expression between PDL2+CD11b+, PDL2+EpCAM+, and PDL2- cells, and observe interesting differences in gene expression confirmed by protein (differential expression of RALDH, IL-27, and IL-10). However, mechanistic studies linking phenotype of DC to function are underdeveloped and preliminary and as such do not significantly add to the message within this manuscript. We plan to pursue this in a more comprehensive way in ongoing studies.

What is the kinetic of PDL2 and CD301b induction on cdc2 after OVA?

Response: We now present a kinetic analysis of PDL2 (and PDL1) expression in LCs and dermal and LN DCs (Figure 6C, new). Interestingly, PDL2 was markedly upregulated by patch placement in epidermal LCs and dermal DCs, but was not increased in OVA+ versus OVA- DCs in the skin. In the draining lymph node, PDL2 upregulation was limited to OVA+ DCs. Changes were observed at 6 h and maintained at 24 h after patch placement. PDL1 was not upregulated by patch placement or by OVA uptake in epidermis or dermis, but was increased in OVA+ DCs as compared to OVA- DCs in lymph nodes.

What happens when PDL2 or CD301b is blocked during OVA uptake? Is Treg generation changed? Are effector T-cells generated?

Response: We do not have sufficient data to make conclusive statements about the role of PDL2 or PDL1 (although if playing a role, our in vitro and in vivo data suggest that the contribution is minor). We have not addressed the role of CD301b. We have added statements in the discussion about the lack of contribution of PDL2 on Th2 priming by PDL2+ DCs, and lack of contribution of PDL2 on Tfh suppression by PDL2+CD301b+ DCs as described in the literature, and that we think that PDL2 is a marker of the cells rather than a mechanism of their impact on T cells.

Are the generated Tregs suppressive ex vivo?

Response: In our previous manuscript (Tordesillas, JACI, 2016), we demonstrated that LAP+ Tregs were suppressive of both T cell proliferation and mast cell activation, and similar in suppressive capacity to sorted CD25+ T cells.

Figure 8

Please show the characterization of the skin myeloid compartment of the SKH1-elite mouse as this is an outbred strain. How does the distribution of DCs look in the skin of hairless mice? Is clustering of DCs visible after OVA application? Or is this orchestrated by a hair follicle dependent process.

Response: We now present data in Supplementary Figure 8 showing the frequency of DC subsets in the skin and lymph nodes of hairless mice. SKH-1 mice showed an increased frequency of LCs in the epidermis compared to Balb/c mice, and a slightly increased cellularity in dermis and lymph node, but the distribution of DC subsets in dermis and lymph nodes was comparable. We have not been able yet to capture the process of antigen uptake by follicle DCs in wild-type mice, so have not examined this in the hairless mice.

Reviewer #2 (Remarks to the Author):

The authors report an interesting site by which CD11b+PDL2+ cDC2 acquire antigen delivered by patch, via the skin hair follicle, to induce LAP+ regulatory T cells. Given the context of their previous work this has important implications for developing systems to induce food tolerance and therefore could be of interest to a broad scientific community.

They provide a thorough characterization of the nature of the DC subsets that acquire and present antigen from the Viaskin patch system. The first two figures establish that the DC subsets that acquire and carry protein antigens from the

skin to draining LNs behave in a similar manner to what has been previously shown for other forms of immunization through the skin (Levin et al J. Invest. Derm 2017 v.137). Further, the nature of the DCs that induce CD4+ T cell activation (CD11b cDC2 and Epcam+ LC) are consistent with prior results in other sites (Dudziak and Nussenzweig et al Science 2007). However, this is a unique system and warrants the characterization performed and the authors execute clean, well-controlled experiments.

The conclusion about the role of LCs in T effector responses has been contentious and potentially depends on the nature of the immunization and the systems used to manipulate LCs. Tordesillas et al demonstrate that LC are indeed CD4+ T cell stimulatory, but like cDC2s, also induce a significant LAP+ regulatory population (Fig. 3). But they are not necessary or sufficient for such responses when deleted using anti-CSF1R and Langerin-DTR(Fig. 4) or conversely when MHCII is removed from non-LC.

Comment from authors: *Although not specifically asked for, we have strengthened our conclusions here by showing that depletion of Langerin+ cells has no impact on the generation of immune tolerance after topical antigen (Figure 4E, new), and secondly have shown that IRF4-dependent DCs are necessary for priming of T cells by the epicutaneous route (Figure 5, new).*

Although their previous work has established the role of the LAP+ Tregs in this patch model, they do not show functionality of the LAP+ T cells induced in vitro by the implicated DCs. Indeed, using their in vitro T cell activation approach they demonstrate Th1, 2 and 17 cytokines (despite referring in the text to a mixed Th1/Th2 response) along with LAP expression, therefore it is unclear what the final in vivo T cell response would actually be. Adoptive transfer experiments might address such questions.

Response: *Although we have not shown that the LAP+ T cells induced by cDC2s in vitro are suppressive, we have shown in previous publications that the LAP+ T cells are suppressive to both T cell proliferation (by in vitro suppression assays) and mast cell activation (by transferring LAP+ T cells to passively sensitized mice), and in that previous manuscript we also showed the multi-cytokine potential of those LAP+ T cells, including production of IL-4 (Tordesillas et al, JACI, 2016). In a second publication, we showed that the functional outcome of topical application of antigen to naïve mice was immune tolerance, mediated by TGF-beta but not by Foxp3+ Tregs (Dunkin, Inflamm Bowel Dis, 2018), consistent with our previous findings in a therapeutic model. Thus the functional outcome of a single topical antigen exposure using a Viaskin patch is TGF-beta-dependent immune tolerance despite the multi-cytokine potential of LAP+ Tregs.*

Despite the rarity of these cells, they go on to elegantly show, using flow cytometry and immunofluorescence, that the relevant DC population, CD11b+ cDC2s, express PDL2 and access epicutaneously applied antigen via the hair follicle. This novel finding suggests how induction of a regulatory rather than inflammatory setting might be established with patching in the absence of adjuvant or skin disruption. Finally, the use of hairless SKH1 mice presents a new approach for studying the function of skin DCs. As such, the characterization of DCs in these mice is warranted (data not shown is indicated in the manuscript). The authors should test the logical implication of the findings presented in Fig. 8: is there a functional difference in the immune response to patched antigens in these mice?

Response: *We now present data on DC subsets in SKH-1 mice in Supplementary Figure 8. We observed that there was a greater frequency of LCs in the epidermal layer and DCs in dermis and lymph node as compared to Balb/c mice, but the distribution of DC subsets was similar. We also provide data demonstrating that the lack of antigen uptake in SKH-1 mice does indeed have functional consequences and these mice do not develop tolerance in response to epicutaneous antigen (Figure 8C, new) .*

Minor comments:

1. The paper is under-referenced. Perhaps this is due to journal restrictions, but certain previous findings are not adequately referred to and/or findings are not put into context of previous work. In addition to some of the references above, the complicated models for LC vs. langerin+ dermal DC depletion using different DTR mice is not referenced (Yao et al 2015 JACI v136 p1387) or discussed.

Response: *this has been added and referenced.*

Also, the conflicting data on the role of cDC2s in Tfh priming is not adequately addressed in the discussion (Kumamoto et al, ref 7 vs. Krishnaswamy et al Sci. Immunol 2017).

Response: This has been expanded.

The role of LAP+ Tregs is not discussed outside of a conclusion from their previous work (Tordesillas et al 2016, ref 12).

Response: We have expanded this slightly to discuss the work of Weiner and colleagues with regard to LAP+ Th3 cells and oral tolerance, but have kept this discussion compact because we do not yet have a satisfactory answer to the mechanism of generation of LAP+ Tregs.

2. Figs. 1-4 and most of the supplementary material do not show statistics.

Response: Where the data is descriptive (identifying subsets acquiring antigen or presenting antigen) rather than testing an intervention, we have not performed statistics and instead show the summary of at least 3 independent replicates per timepoint or condition. Where we have performed interventions, we have used groups powered for statistical analysis.

3. The Viaskin patch system is not described and therefore the background on why such a system might promote Tregs is not clear.

Response: We have expanded the discussion of the Viaskin patch.

4. In the discussion, the authors reference PDL2+ Cd11b+ DCs driving Th2 but not IgE responses, yet Gao et al (ref 5) demonstrated impaired IgE in IRF4-/- mice.

Response: We have amended this point in the discussion and expanded on the differing results between different model systems.

Reviewer #3 (Remarks to the Author):

COMMENTS, QUESTIONS AND SUGGESTIONS:

1. The authors show uptake of OVA by LCs and they show that OVA can be detected in the draining lymph nodes. Probably not unexpectedly, the percentage of LCs that takes up OVA in situ is quite low (between 6 and 16% of all LCs). So are the corresponding percentages of OVA-carrying DCs in the draining LNs. Some points regarding these observations may need a little more documentation or reporting of observations / data:

1A. The Viaskin patch does not contain an adjuvant. What is the migration stimulus for the LCs (and dermal DCs)? Is the occlusive milieu under the "lid" of the patch inflammatory? LPS-containing OVA? A simple H&E histology may give some clues if there is inflammation - influx of lymphocytes and / or neutrophils.

Response: To examine the impact of the patch on the skin milieu, we have performed RT-PCR for cytokine expression (Figure S6, new). Placement of the patch resulted in transient upregulation of a number of cytokines and chemokines, indicating that skin occlusion did result in mild inflammation.

1B. For the sake of completeness, the authors should report whether epidermal keratinocytes also took up some OVA. In the present figure they were gated away. (I realize, that this information is contained in the authors' previous JI paper from 2011 - Dioszeghy et al. This paper is not cited here, though. Please cite it.)

Response: We did not have an epithelial-specific marker, although CD45-negative cells were positive for fluorescence associated with the antigen. There was a global shift in fluorescence in these CD45 negative cells, and it was not convincing that this was a true active uptake mechanism. We have not discussed this in the paper. We have included the citation (Dioszeghy) with discussion, although it is not clear that the surface immunofluorescence that was shown in the microscopy of that paper was surface fluorescence or intracellular uptake into the keratinocytes.

1C. Connected to the previous point, it would be important to know whether LCs indeed emigrate from the epidermis in substantial, measurable numbers. This could be easily tested by either looking at MHCII- or Langerin-stained epidermal sheets or by analyzing epidermal cell suspensions for their percentages of MHCII or langerin+ cells. Sheets may be more straightforward. If emigration is observed, how would it compare to what is published for explant cultures (where the vast majority of LCs leave the epidermis over a period of about 3 days) or for CHS experiments, where the numerical reduction of LCs can also be readily observed and quantified. (Again, I realize that this was looked at in the not cited previous JI paper from 2011 - Dioszeghy et al. - Fig.4C. Please mention.)

Response: We have looked at OVA-bearing LCs, and their absence in CCR7-/- mice, demonstrating that LCs do migrate to the lymph node from the skin. As we have presented data here showing that LCs are not necessary for either presentation in vivo or the induction of tolerance, we have not investigated presentation by these cells in greater detail.

1D. The authors show that OVA+ cells in the LNs reside in the MHC-high/CD11c fraction. How are the percentages of OVA+ cells in this fraction? In Fig.2A the authors look at total DCs where percentages are low. In the migratory fraction the percentages should be markedly higher. This information should be retrieved from the available data and presented.

Response: We now present data showing the percentage of OVA+ cells in the migratory DC fraction, the resident DC fraction, and the (CD64+) macrophage population (Figure 2D, new). OVA+ cells were not observed in the resident DC population but were observed in the CD64+ macrophage population.

1E. Langerhans cells in the lymph nodes. In Fig.2E it becomes apparent that a population of EpCAM+ cells, i.e. LCs carry OVA. What about the inverse? Do all LCs in the node carry OVA? Probably not, since even in the epidermis only a small part of LCs take up the OVA. Is it then correct to read Figure 2F such that of 20,000 EpCam+ cells only about 50 carry OVA 48hrs after application? This would mean only 0.25%? Correct? If so, please state in an unequivocal and affirmative way. If not, please try to word it such that misunderstandings will be avoided. I think it is important to report such numerical relations / proportions.

Response: We have modified the figure so that the frequency of OVA+ cells is now expressed as a percentage of the total subset. Thus, at 72 hours the frequency of OVA+ EpCAM+ DCs in the lymph node is approximately 2% of all EpCAM+ DCs. Only a small fraction of EpCAM+ DCs (or CD11b+ DCs) in the lymph node carry detectable antigen.

2. Referring to data from Figure 6 - PDL2 and CD301b. These markers are expressed at higher levels in the migratory fraction of skin draining nodes (Fig.6B). Since this paper emphasizes the epicutaneous aspect (and also because Gao-Medzhitov in ref.7 show that PDL2/CD301b+ cells contain LCs and dermal DCs / and not least also because DVB Technologies emphasizes Langerhans cells in their website description) it would be interesting and relevant to explore this point more in detail. Two suggestions: (A) In Fig.1 the authors show some phenotype of skin cells after 48 hrs under the VIASKIN patch - both epidermal LCs and dermal DCs. Do these DCs turn PDL2 and/or CD301b positive? Both fractions - the OVA-positive and the OVA-negative fraction? (B) This may be technically not possible? Place the VIASKIN patch onto an explant culture and look at "crawl-out" DCs. In this context: A previous paper by Pieter Leenen's group has shown up-regulation of CD301 on LCs upon explant culture. Back then, CD301a versus CD301b could not be distinguished, though (J.Leukocyt.Biol. 80:839, 2006).

Response: We have focused more extensively on PDL2 expression on cells in the epidermis and dermis as they acquire antigen by tracking PDL2 expression on OVA+ and OVA- DCs. Placement of the patch upregulates PDL2 on epidermal and dermal DCs, independent of antigen uptake (Figure 6C, new). This is likely related to the mild inflammatory response to patch placement (Figure S6, new). A major difference between OVA+ and OVA- DCs is not observed until the draining lymph node, when there is a striking difference between PDL2 expression in OVA+ versus OVA- migratory DCs (Figure 6C). PDL1 is not regulated in the skin, and is elevated in OVA+ cells as compared to OVA- cells in the lymph node.

3. Referring to data from Figure 6 and 7 - combined. In 6A the upregulation of PDL2 is shown. In 7A the proliferation of DO11 cells induced by sorted DCs from the draining nodes is depicted. PDL2-negative cells do not induce proliferation. Why? Do they not take up the OVA? This could be directly assessed using the Alexa647-OVA patch.

Response: PDL2 is highly upregulated in antigen-bearing DCs after leaving the skin. OVA+ DCs are uniformly PDL2 high and therefore the PDL2-negative fraction does not contain OVA, as the reviewer guesses. This is now stated clearly in the text, and shown in Figure 6D.

4. Significance asterisks are only from Figure 5 onwards. Does this mean that none of the important data on antigen uptake and carriage in Figure 1 - 4 are significant? Please clarify in an unequivocal way for each figure.

Response: Statistics were not performed for descriptive experiments where we repeated experiments a minimum of 3 times, and number of replicates are indicated in the figure legends. We performed statistical analysis using appropriately powered group sizes after interventions such as cell depletion or tolerance induction.

5. How many times was the experiment in Suppl. Fig 1 (CCR7-/- mice) done?

Response: 3 mice per group were used for this experiment showing reduced T cell proliferation in CCR7^{-/-} mice. The flow plots are representative.

6. Suppl. Figure 2. Please clarify: Is "B" the summary of all 4 experiments, one representative of which is depicted in "A"? The legend is somewhat confusing in this regard. This must be stated because %CFSE of CD11⁺ and EpCAM⁺ cells in "A" appear quite similar, in "B", however, they look very different.

Response: B is the summary of all experiments. The representative plots (without antibody treatment to deplete LCs) show 48.7% proliferation after culture with EpCAM⁺ DCs (range 48.7-76.3) versus 34.3% after culture with CD11b (range 11.4 – 38.6).

7. Figure 8A. Beautiful pictures - good resolution! Yet, not sure how to interpret them. I cannot really see the grey for the OVA (A647-OVA, I assume). Or is it the light blueish in the high magnifications? If so, a few little arrows may help. Moreover, in the anti-CSFR1-treated mice langerin is absent but there are grey DCs in the epidermis? This is confusing, since LCs are / should be depleted.

Response: Although OVA is present on the skin surface, this does not appear to be intracellular or internalized in the epidermis. Despite washing of the skin prior to preparation for microscopy, some surface antigen remains, this is not in LCs. We did not capture OVA uptake by these DCs in these photos, and use the photos to show that both LCs and non-LC DCs can extend dendrites across the follicle epithelium. The finding that hairless mice have impaired antigen uptake (by flow cytometry) and impaired tolerance induction (new data: Figure 8C) further supports the concept that antigen acquisition occurs through the follicle. To avoid confusion, we have removed the "OVA" label on the micrographs.

8. The authors may want to briefly touch upon the "change of paradigm" between their 2011 JI paper, where LCs still were the suspects (although without experiments directly showing this) and this manuscript, that rules out LCs as mediators of the clinically observed effects of EPIT.

Response: We have included this in the discussion. However, we cannot conclude that LCs are dispensable in sensitized mice, as our experiments here address the role of DC subsets in naïve mice during the induction of primary tolerance, and our collaborators at DBV have focused primarily on mechanistic studies in sensitized mice that more appropriately model the response during allergen immunotherapy.

MINOR POINTS

9. M&M. Please specify which langerin-DTR mice were used. The ones generated by Kissenpfennig/Malissen or those by Clausen/Bennett or were they made "in house"?

This is now stated (Kissenpfennig/Malissen).

10. The authors write: "In vitro loading of all DC subsets with OVA peptide as positive control led to proliferation (not shown)." This important control should be shown in a supplementary figure.

This is now included as Supp Table 1, with the important modifier that we did not observe proliferation with the CD8 subset loaded ex vivo with peptide. This has been noted in the text.

11. Figure 2E. Perhaps the authors may want to give some words of explanation for the gate in the upper left panel that looks a little strange at first glance. It was chosen that way in order to include both migratory LNDCs (MHCII-high) and resident LNDCs (MHCII-lower). Correct?

We have revised and now also included a gating figure (Figure S1) to make it clearer how we gate for total and migratory DCs. For antigen uptake we wanted to also examine if there was acquisition by resident DCs.

12. Fig.4. What is the difference between panels "C" (% CFSE-low DO11 cells) and "D" (% OVA-specific CD4 cells). CFSE seems clear to me. But what about "D"? Is the legend wrong in that "D" would be the same as "F", i.e., pentamer-positive endogenous CD4 cells? This would make sense. Please get this right.

Response: C (CFSE-low) includes only the fraction of the OVA-specific T cells that are proliferating, while D includes both CFSE high and CFSE low cells. These are all transferred DO11.10 (B) or OT-II cells (D). We observed a slightly lower accumulation of T cells in the draining lymph node after LC depletion, while the rate of proliferation of these cells was not affected. Because of the slightly lower T cell accumulation, we wanted to show both data representations. However, we

now show that LC depletion has no impact on the functional readout of immune tolerance (Figure 4E).

13. The authors write: "Despite the complete depletion of LCs in the skin, there was only a 50% reduction in presentation of antigen by EpCAM+ DCs in the draining lymph node (Supp.Fig 2), suggesting that this population could also include langerin+ dermal DCs that upregulate EpCAM after migration to the LN." Is there any published evidence that the dermal cDC1s (CD103+/langerin+) upregulate EpCAM? If so, please cite. If not please state that this is a speculation only. Either way is fine.

We have modified this statement to not discuss upregulation, as there is published description of EpCAM (at a lower level than LCs) on langerin+ dermal DCs, and have cited this literature.

14. Suppl.Fig.3 - FACS analyses. Please show a representative dot plot for the PD-L2 staining.

We have now included PDL2 staining in Figure 6.

15. Legend to Fig.7B. By "staining control" the authors mean "isotype control", I guess. Correct? Please state explicitly.

This is a fluorescence minus one (FMO) control, and this notation has been added to the legend.

16. Adding the word "outbred" to the SHK1 Elite hairless mice in M&M might preempt the question for functional T cell assays in these mice.

This has been added.

Reviewers' comments:

Reviewer #1 (Remarks to the Author):

No further comments

Reviewer #2 (Remarks to the Author):

In general the authors have adequately addressed the concerns initially raised and strengthened the paper with new data and written sections; a few exceptions are listed below along with a concern from the new data.

1. The answer to my minor comment #2 seems unusual – statistics are routinely used to compare many types of data in order to let the reader know how robust the differences are between compared data (even if pooled from multiple experiments). How do we know if the cells types indicated in Fig 2 are different with treatment, over time, etc.? And multiple interventions are used in Figs. 1-4 yet stats are not indicated. For example, OVA + anti CSF1R in Fig 4A-B, DT in Fig 4C. Even when trying to compare different APCs in Fig 3B in terms of T cell activation, are there real differences in ability to activate? The authors seem to conclude so. Reviewer 3 raised this concern as well. And now I have the same issue with the new Fig 8B-C. It lacks statistical analysis and the figure legend for 8C is very general without sufficient detail.

2. In the new Fig 5, the authors appropriately add analyses using DC-specific IRF4-deficient mice to eliminate cDC2s. However, this mouse has serious problems with off-target deletion (see Schitzler et al, Immunity 2013 p970). The authors should add whether they screened their mice for significant T cell deletion of IRF-4 using the GFP reporter that is part of the construct, as such deletion could impact the results presented in Fig. 5C.

Reviewer #3 (Remarks to the Author):

I PICKED UP SOME MINOR POINTS:

1. Suppl.Fig 1B - the "DC contour plot" is lacking the x-axis label.

2. Forgot to refer to Figure 8C in the text.

3. The authors write in the Results: "CD11c-Cre x IRF4^{fl/fl} showed a marked absence of CD11b+EpCAM- DCs in the migratory DC compartment and a markedly suppressed proliferative response of transferred OVA-specific T cells (Fig 5A, B)." I think "suppressed" is not the correct word. It insinuates something actively suppressing this response. This is not the case, however. The proliferative response is less, simply because the responsible cells (IRF-4 DCs) are missing. Therefore, "markedly reduced" would be more appropriate.

AND ONE "NOT SO MINOR" POINT

3. Figure 8C. This is the only experiment, that really looks at classical tolerance: BALB/C versus hairless = hair-follicle-less mice. Please describe this assay still a bit more in detail. A small paragraph was introduced into the revised version. Now it reads: "Mice received a Viaskin® patch loaded with 100 µg of OVA as above for 48h, or as control the skin was prepared in the same manner but no patch applied. Seven days after patch application, micewere injected in each hock with 50 µg of OVA in Complete Freund's Adjuvant 354 (Sigma Aldrich, St. Louis, MO), and after

another 14 days boosted with OVA in incomplete Freund's Adjuvant (Sigma Aldrich)." And then?
How long after the boost were LNs taken and IFN-gamma measured?

We would like to thank the reviewers for their helpful suggestions. We have addressed these remaining concerns, and point-by-point responses are provided below.

Reviewer #2 (Remarks to the Author):

In general the authors have adequately addressed the concerns initially raised and strengthened the paper with new data and written sections; a few exceptions are listed below along with a concern from the new data.

1. The answer to my minor comment #2 seems unusual – statistics are routinely used to compare many types of data in order to let the reader know how robust the differences are between compared data (even if pooled from multiple experiments). How do we know if the cells types indicated in Fig 2 are different with treatment, over time, etc.? And multiple interventions are used in Figs. 1-4 yet stats are not indicated. For example, OVA + anti CSF1R in Fig 4A-B, DT in Fig 4C. Even when trying to compare different APCs in Fig 3B in terms of T cell activation, are there real differences in ability to activate? The authors seem to conclude so. Reviewer 3 raised this concern as well. And now I have the same issue with the new Fig 8B-C. It lacks statistical analysis and the figure legend for 8C is very general without sufficient detail.

Response: We now provide results of statistical analysis on all figures throughout the manuscript.

2. In the new Fig 5, the authors appropriately add analyses using DC-specific IRF4-deficient mice to eliminate cDC2s. However, this mouse has serious problems with off-target deletion (see Schlitzer et al, Immunity 2013 p970). The authors should add whether they screened their mice for significant T cell deletion of IRF-4 using the GFP reporter that is part of the construct, as such deletion could impact the results presented in Fig. 5C.

Response: We do indeed observe off-target GFP expression in B cells and T cells as previously described. We also examined IRF4 expression by intracellular immunostaining. Activation of CD4+ T cells with CD3/CD28 resulted in robust upregulation of IRF4, although at a lower level than wild-type mice, while DCs had completely abolished IRF4 expression. Although we were aware that these mice had issues with off-targeted deletion, we felt that the use of wild-type OT-II cells as our read-out would limit that concern. However, we have added lines to the text to indicate off-target deletion does occur in these mice, and this is a limitation of the model system.

Reviewer #3 (Remarks to the Author):

I PICKED UP SOME MINOR POINTS:

1. Suppl.Fig 1B - the “DC contour plot” is lacking the x-axis label.

Response: We have fixed this oversight.

2. Forgot to refer to Figure 8C in the text.

Response: This is now cited in the text.

3. The authors write in the Results: “CD11c-Cre x IRF4^{fl/fl} showed a marked absence of CD11b+EpCAM⁻ DCs in the migratory DC compartment and a markedly suppressed proliferative response of transferred OVA-specific T cells (Fig 5A, B).” I think “suppressed” is not the correct word. It insinuates something actively suppressing this response. This is not the case, however. The proliferative response is less, simply because the responsible cells (IRF-4 DCs) are missing. Therefore, “markedly reduced” would be more appropriate.

Response: We agree, and have changed the wording.

AND ONE “NOT SO MINOR” POINT

3. Figure 8C. This is the only experiment, that really looks at classical tolerance: BALB/C versus hairless = hair-follicle-less mice. Please describe this assay still a bit more in detail. A small paragraph was introduced into the revised version. Now it reads: “Mice received a Viaskin[®] patch loaded with 100 µg of OVA as above for 48h, or as control the skin was prepared in the same manner but no patch applied. Seven days after patch application, micewere injected in each hock with 50 µg of OVA in Complete Freund's Adjuvant 354 (Sigma Aldrich, St. Louis, MO), and after another 14 days boosted with OVA in incomplete Freund's Adjuvant (Sigma Aldrich).” And then? How long after the boost were LNs taken and IFN-gamma measured?

Response: We have increased the detail in the methods to more effectively describe the tolerance experiments.

REVIEWERS' COMMENTS:

Reviewer #2 (Remarks to the Author):

The authors have adequately addressed all of my concerns

Reviewer #3 (Remarks to the Author):

none